# Are Large Language Models Ready for Multi-Turn Tabular Data Analysis?

Jinyang Li [* 1]   Nan Huo [* 1]   Yan Gao [2]   Jiayi Shi [3]   Yingxiu Zhao [4]   Ge Qu [1]   Bowen Qin [5]   Yurong Wu [2]
Xiaodong Li [6]   Chenhao Ma [3]   Jian-Guang Lou [2]   Reynold Cheng [1]

## Abstract

Conversational Tabular Data Analysis, a collaboration between humans and machines, enables real-time data exploration for informed decision-making. The challenges and costs of collecting realistic conversational logs for tabular data analysis hinder comprehensive quantitative evaluation of Large Language Models (LLMs) in this task. To mitigate this issue, we introduce **CoTA**, a new benchmark to evaluate LLMs on conversational data analysis. CoTA contains 1013 conversations, covering 4 practical scenarios: NORMAL, ACTION, PRIVATE, and PRIVATE ACTION. Notably, CoTA is constructed by a multi-agent environment, **DECISION COMPANY**. This environment ensures efficiency and scalability of generating new conversational data. Our comprehensive study, conducted by data analysis experts, demonstrates that DECISION COMPANY is capable of producing diverse and high-quality data, laying the groundwork for efficient data annotation. We evaluate popular and advanced LLMs in CoTA, which highlights the challenges of conversational tabular data analysis. Furthermore, we propose **A**daptive **C**onversation **R**eflection (**ACR**), a self-generated reflection strategy that guides LLMs to *learn from successful histories*. Experiments demonstrate that ACR can evolve LLMs into effective conversational tabular data analysis agents, achieving a relative performance improvement of up to 35.14%. Code can be found at https://tapilot-crossing.github.io/

## 1. Introduction

The exponential growth of big data calls for accessible data analysis techniques that cater to a wide range of applications, such as healthcare, games, and entertainment (Khanbabaei et al., 2018; Han et al., 2011; Fayyad et al., 1996). Recently, the development of LLM agents (Liu et al., 2024b; Xu et al., 2024; Zeng et al., 2024; Xu et al., 2023; Deng et al., 2024; Si et al., 2023) has attracted a lot of attention. They are capable of understanding natural language queries, as well as generating codes for data manipulation and visualization, through reasoning (Huang & Chang, 2023; Wei et al., 2022; Wang et al., 2024a) and tool calls (Li et al., 2023b; Huang et al., 2024b; Qin et al., 2024). Among the vast types of data available, tabular data stands out as one of the most prevalent and interpretable formats organized by rows and columns (Hu et al., 2024; Wu et al., 2024; Liu et al., 2024a).

Tabular data analysis agents (Li et al., 2024b; Zha et al., 2023; Zhang et al., 2023a) provide automatic workflow based on user queries. However, the dynamic and uncertain nature of real-world analysis hinders effective human-agent conversation (De Vries et al., 2020; Yan et al., 2023; Wang et al., 2024b), thus users may need to adjust their analysis strategies based on intermediate results (Yan et al., 2023; Yao et al., 2020). For example, in Figure 1, the `notable opponents` could refer to a variety of interpretations, such as the opponents with the highest wins, or the most frequent opponents. Towards this end, a comprehensive benchmark is indispensable for gauging their capability in conversational user engagement within data analysis scenarios.

In this paper, we introduce **CO**nversational **T**abular data **A**nalysis (**CoTA**), a new benchmark for evaluating LLM agents in conversational tabular data analysis tasks. CoTA is designed to simulate real-world data analysis scenarios, where users converse with LLM agents to generate codes for data exploration and decision making. It includes 1013 user-machine conversations with 1162 user intents, spanning four practical scenarios, as shown in Figure 1: **1) Normal** mode refers to the scenario where all questions and user requirements are explicit, agents can answer questions by referring only to table contents and dialog histories. This would evaluate fundamental capabilities of agents in handling data analysis tasks; **2) Action** mode represents that agents must

---

[*]Equal contribution  [1]School of Computing and Data Science, The University of Hong Kong [2]Microsoft [3]The Chinese University of Hong Kong, Shenzhen [4]Alibaba Group [5]Beijing Academy of Artificial Intelligence [6]Xiamen University. Correspondence to: Reynold Cheng <ckcheng@cs.hku.hk>, Chenhao Ma <machenhao@cuhk.edu.cn>.

*Proceedings of the 42nd International Conference on Machine Learning*, Vancouver, Canada. PMLR 267, 2025. Copyright 2025 by the author(s).

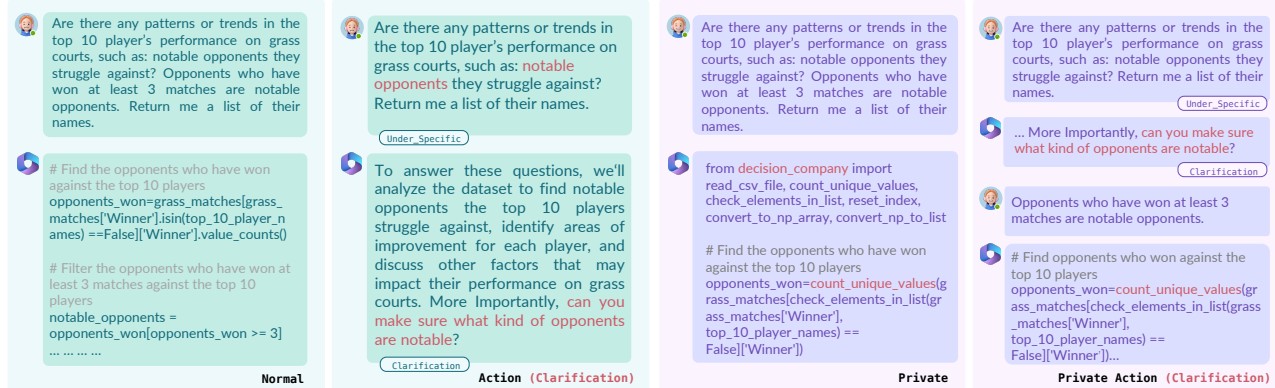

*Figure 1.* An overview of the four conversation modes in COTA, illustrated by relevant aspects of the associated codes or actions.

infer diverse user intents first to deliver satisfactory results. For example, they need to interpret ambiguous terms such as `notable opponents` by asking questions and generate appropriate responses based on user clarification. This tests their ability to respond to complex and dynamic user queries during conversations; **3) Private** mode is designed to examine the true semantic parsing capability of agents when encountering unseen packages provided by users (Zan et al., 2022); and **4) Private Action** mode unifies the challenges of Private and Action modes, more closely reflecting real-world data analysis. Answer types can be summarized into two categories: **1) Code Generation**, which can test whether the agent can correctly interpret the user query and generate the corresponding code for data analysis, and **2) Multiple-Choice questions**, which can evaluate the ability of agents to understand the returned results being executed codes and provide users with appropriate insights.

Constructing high-quality conversational data analysis datasets poses significant challenges. First, obtaining realistic human–machine conversational logs for data analysis can be difficult due to proprietary restrictions and privacy concerns (Sun et al., 2022; Choi et al., 2019). Second, the conventional approach of using crowdsourcing for dataset construction, particularly for complex, high-quality conversational tasks, demands substantial human expertise, making it both time-consuming and costly (Yu et al., 2019a; Li et al., 2023a; Guo et al., 2021; Li et al., 2024d; Zhang et al., 2023c). Furthermore, such methods are prone to data leakage risks. To address these challenges, we introduce a novel multi-agent sandbox, **DECISION COMPANY**, enabling the efficient creation of COTA through human–sandbox collaboration under stringent execution-based evaluation script comprising test cases. In this simulated sandbox, 4 GPT-4 agents interact with each other under continuous human expert supervision to simulate realistic data analysis tasks while maintaining high data quality. The reliability and potential biases of such human-sandbox collaborative ap-

proach are rigorously evaluated through 10 real data analysis experts outside annotations. The result shows that DECISION COMPANY can scale and maintain data quality.

We evaluate the popular advanced LLMs and LLM agents on COTA. The results underscore the challenges of conversational data analysis and fuel the need for more advanced LLM agents that can handle diverse user intents and feedback. To further evolve the LLMs towards effective conversational data analysis agents, we propose **A**daptive **C**onversation **R**eflection (**ACR**), which guides LLM agents to ***learn from successful history*** via self-generated pseudo logic reflection. Our experiments demonstrate that **ACR** can significantly enhance the performance of LLMs, in which Claude-3.5-Sonnet can gain relative improvement of **35.14%** compared to its model base, offering an insight into how to improve LLM agents in conversational tabular data analysis.

## 2. Preliminaries

**Task Formulation.** Conversational tabular data analysis with LLM agents involves a sequence of user-agent turns, $[(u_1, a_1), (u_2, a_2), \ldots, (u_n, a_n)]$, where each turn $(u, a)$ consists of a user query $u$ and an agent response $a$. Queries can be instructions or feedback, while responses can be code snippets or selected answers. Dialogs start with $u_1$ and end with $a_n$. Given the current user query $u_t$, all previous user-agent history $H$ from turn 1 to $t-1$, and sampled table contents $T$, the agent should act, such as asking for clarification, and generate an answer $a_t = f_\theta(u_t, H, T)$, where $f_\theta$ refers to the agent built based on LLMs with model weights $\theta$. This setup allows COTA to evaluate conversational agent performance in a static and systematic manner.

**Action Types.** We identify 6 common actions during conversations, each serving as a specific evaluation mode in COTA. The actions in COTA include `Update_Code`, which addresses user requests for bug fixes or refinements;

`Fast_Fail`, which alerts users to insufficient data or factual errors; and `Clarification`, where agents seek additional information for under-specified queries. To reduce user impatience and long dialogue issues, `Best_Guess` allows agents to make assumptions based on data, domain knowledge, and commonsense, though it risks incorrect guesses. The `Plot_QA` action helps users understand plot-derived insights, while `Insight_Mining` involves summarizing executed results to aid in decision-making, evolving agents into comprehensive data analysis tools. Detailed examples and interpretations are in Appendix J, and evaluation methods for each mode are in Appendix L.

## 3. The COTA Construction

### 3.1. Sandbox Construction

The construction of COTA is mainly based on the AI Agent Sandbox, DECISION COMPANY, as depicted in Figure 2.

**Data Acquisition & Preprocessing.** The first step in the construction of COTA is the acquisition and preprocessing of data. We collect open-source tables from Kaggle, a popular data science platform. These tables cover 18 analysis topics under 5 common domains, namely `ATP Tennis`, `Credit Card`, `Fast Food`, `Laptop Price`, and `Melbourne Housing`, as detailed in Appendix B.3. Given any of the tables, Administrator Agent will generate column meanings and value illustrations.

**Client Persona Generation.** The construction of COTA proceeds to the generation of client personas. These personas with specific tasks and topics related to the data are created by the Administrator Agent. Each persona is defined by a `Name`, `Location`, `Job`, and `Background` with a diverse range of interests and backgrounds.

**Simulation of Analysis Scenarios.** Then, the Administrator Agent "interviews" each Client Agent to obtain their `Scenario` description, including `Scenario Name` and the `Goal`. In preliminary trials, we observe that Client Agents often proposed very uniform topics, which limited the diversity of the dialogue. To address this, each Client Agent is instructed to propose 3 scenarios, from which human annotators selected most appropriate one scenario based on its diversity and coherence with the tabular data. For instance, in B.3 of Figure 2, the scenario `Court Condition Impact` was chosen because `Player Performance Analysis` was less diverse since multiple Clients tend to generate it as the top choice, and `Sponsor Attraction` required extensive information beyond the table contents, leading to many unanswerable questions.

**Plan Discussion.** In this process, the Data Scientist Agent engages with the Client Agent to convert the requirements of client into a series of specific data analysis questions with well-defined conditions. Each question is provided by an expected result type, such as dataframes, lists, or various plot types, which helps reduce question ambiguity and ease the pressure on evaluation metrics (Yin et al., 2023; He et al., 2024; Zhang et al., 2023b). The dialogue between the agents further refines the questions with specific conditions. For example, as depicted in Figure 2 B.4, the client `Garcia`'s question could be further elaborated on the basis of his following responses, making all questions more answerable. In particular, Agent `Garcia`, fully cognizant of his persona created in B.2, adds the condition `grass`, reflecting his `London` location. This implies that the role-playing aspect of the agent can be instrumental in generating a wider range of questions that are both diverse and reasonable (Li et al., 2024a; Park et al., 2023).

**Conversation Log Generation.** Following the plan discussion, the conversation simulation phase begins. Here, the AI Chatbot Agent takes the lead, executing the data analysis plan agreed on during the previous stage. The AI Chatbot Agent converses with the Data Scientist Agent to answer a series of questions defined in the plan by generating codes and analyzing returned results.

### 3.2. Human-Sandbox Annotation

While the DECISION COMPANY can generate a wealth of data analysis conversations in a zero-shot prompting manner, human supervision is indispensable to ensure the quality of the data set annotation (Lu et al., 2023; Zhuo et al., 2025). Therefore we engaged two groups of experts: 1) 6 annotators who are also authors, each with over 10 years of experience in data analysis. 2) a team of 3 more experienced expert data scientists for final data collection decision.

**Annotation Phase.** Each annotator begins by Simulation of Analysis Scenarios in DECISION COMPANY to select scenarios and continue process until arriving Conversation Log Generation for annotation. An illustrative starting prompt is provided in Figure 17, Appendix Q. Beginning with this prompt, the AI Chatbot generates code in response to various analysis questions. Each annotator then executes the generated code, verifies its outputs for correctness, fixes any bugs, and confirms that the results align with the anticipated output types, ensuring that the code accurately addresses the posed questions. This finalized code is referenced as the *ground-truth* code for subsequent benchmarking. Additionally, the annotator generate evaluation scripts (`eval.py`) consisting of a series test cases for testing model-generated code. After running these tests, the annotator executes the code and feed the results back to the AI Chatbot, which generates free-form text analysis statements (e.g., `"The credit card application rate of people with 4-year employment is 73.5% higher than`

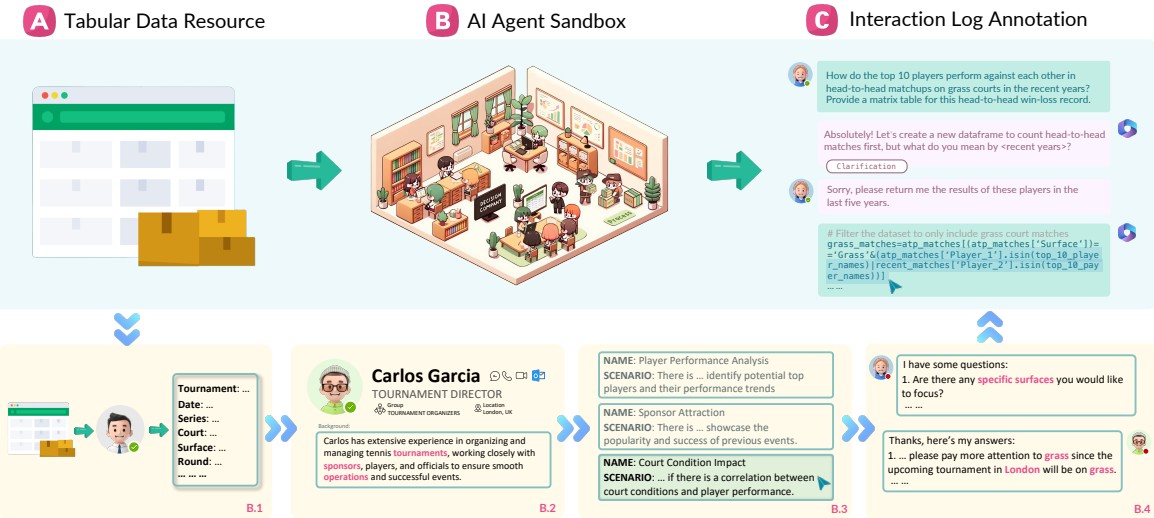

*Figure 2.* The construction pipeline of CoTA by the AI Agent Sandbox DECISION COMPANY. ⬈ denotes human intervention

those with no employment"). Each annotator reviews and corrects these statements, converting them into multiple-choice question formats for more reliable and objective evaluation. All corrected code and free-form text analysis are stored as static user-AI conversation history, for reference when annotating the following turns. Each conversation is presented as series of clean, noise-free turns. Annotators then inject meaningful action types, shown in Section 2, drawing on established dialogue research and their practical experience in data analysis. For *private library* code annotations, each annotator will follow three steps to annotate: 1) summarizing packages from original code; 2) refactoring and converting them to user customized functions with light human supervision; 3) regenerating codes via customized functions with human calibration. Further details are provided in Appendix Q.

**Validation Phase.** Once the conversation annotations are complete, annotators swap their annotated data for cross-validation. The validation process involves three key steps: (1) adding more test cases to the evaluation scripts to check the code; (2) assuming the role of a "red team" to intentionally introduce errors into the code, then confirming the evaluation scripts correctly flag these errors; and (3) critically judging the suitability of the chosen scenarios and action types. Annotators resolve any identified errors or disagreements through discussion. If they cannot reach consensus, the expert team steps in to refine or, if necessary, eliminate the disputed items.

## 4. Data Statistics & Metrics

### 4.1. Dataset Statistics

Figure 1 provides key statistics for our dataset, while Table 2 offers a comparison between CoTA and other datasets related to data analysis. To ensure a fair comparison

*Table 1.* Data characteristics

| STATISTIC | NUMBER |
|---|---|
| **Total conversations** | 1013 |
| ▸ clear conversations | 280 |
| ▸ action conversations | 478 |
| ▸ private lib. conversations | 206 |
| ▸ private act. conversations | 49 |
| ▸ # of private lib functions | 137 |
| **Answer Types** | |
| ▸ # of code generation answers | 590 |
| ▸ # of multi-choice answers | 423 |
| **Quality & Cost** | |
| ▸ inter-agreement | 92.78 |
| ▸ AVG # of turns | 14.15 |

regarding question and code length, we utilize tiktoken to compute the number of tokens for each dataset. As shown in Table 2, CoTA includes comprehensive evaluation settings across private library, multi-turn, and multi-modal conversations. Besides, the complexity of this dataset, reflected by the long questions and their associated code snippets, is amplified by the inclusion of multi-intent queries. These queries, encapsulating multiple intents within a single question, require a versatile array of computational strategies for effective handling. For example, the query, "Please provide histogram plots and mean for employment status of credit card applicants." demands both data visualization and statistical evaluation. Finally, CoTA contains 1013 data analysis conversations. The inter-agreement of 92.78 for initial cross-validation promises the high quality of the dataset.

*Table 2.* Comparison of CoTA and other data analysis datasets. The first 5 datasets are single-turn data analysis sets featuring both SQL and Python codes. The following 3 benchmarks are multi-turn or conversational data analysis datasets. CoTA represents a challenging dataset in data analysis with more comprehensive settings. 🐍 represents that the end code is Python. 🗄 means the target code is SQL.

| Dataset | # Q ǀ # Intents | # Toks. / Q | # Toks. / Code | Code Type | Analysis | Multi-Turn | Private Lib | Multi-modal | Evaluation |
|---|---|---|---|---|---|---|---|---|---|
| HumanEval (Chen et al., 2021) | 164 ǀ 164 | 60.9 | 24.4 | 🐍 | ❌ | ❌ | ❌ | ❌ | Test Cases |
| MBPP (Austin et al., 2021) | 974 ǀ 974 | 14.5 | 24.2 | 🐍 | ❌ | ❌ | ❌ | ❌ | Test Cases |
| Spider (Yu et al., 2018) | 1034 ǀ 1034 | 12.4 | 18.3 | 🗄 | ❌ | ❌ | ❌ | ❌ | Acc + EM |
| BIRD (Li et al., 2023a) | **1534 ǀ 1534** | 14.5 | 49.6 | 🗄 | ❌ | ❌ | ❌ | ❌ | Acc + VES |
| DS-1000 (Lai et al., 2023) | 1000 ǀ 1000 | **282.4** | 42.1 | 🐍 | ❌ | ❌ | ❌ | ✅ | Test Cases + SFC |
| SparC (Yu et al., 2019b) | 1203 ǀ 1203 | 9.4 | 26.3 | 🗄 | ❌ | ✅ | ❌ | ❌ | Acc |
| CoSQL (Yu et al., 2019a) | 1008 ǀ 1008 | 13.1 | 31.4 | 🗄 | ❌ | ✅ | ❌ | ❌ | Acc |
| ARCADE (Yin et al., 2023) | 1066 ǀ 1066 | 19.2 | 48.2 | 🐍 | ❌ | ✅ | ❌ | ❌ | Acc + Fuzzy |
| **CoTA** | 1013 ǀ 1162 | 207.5 | **164.7** | 🐍 | ✅ | ✅ | ✅ | ✅ | Acc + AccR |

*Table 3.* Acceptance ratio of human evaluation on general metrics of the dataset quality, **the higher the better**. The table reports the percentage of samples considered qualified or being accepted for each metric.

| Annotation | Conversation | | | | | Eval Scripts | |
|---|---|---|---|---|---|---|---|
| | Conversation Coherence | Scenarios Diversity and Reasonableness | Conversation Topic Coherence | Ethics and Bias Representation | Conversation Naturalness | Evaluation Scripts Quality | Evaluation Scripts Scalability |
| w/o human | 0.19 | 0.46 | 0.17 | 0.41 | 0.67 | - | - |
| w/ human | **0.97** | **0.96** | **0.93** | **1.00** | **0.95** | **0.98** | **0.94** |

## 4.2. Evaluation Metrics

**Accuracy (Acc).** Acc is a metric that evaluates the ability of agents in generating codes that execute correctly or answer multiple-choice questions accurately. It is defined as the proportion of instances whose predicted outputs match the ground-truth output, examined by evaluation scripts. For a given dataset with $N$ instances, where $C_i$ is the expected outcome (either execution result or correct answer) and $\hat{C}_i$ is the predicted outputs for the $i^{th}$ instance, Acc is calculated as follows:

$$\text{Acc} = \frac{1}{N} \sum_{i=1}^{N} \mathbf{I}(C_i = \hat{C}_i), \tag{1}$$

where $\mathbf{I}$ is an indicator function that returns 1 if $C_i = \hat{C}_i$, and 0 otherwise.

**Accuracy with Private Library Recall (AccR).** Recognizing the importance of accurately leveraging specific user-defined libraries in code generation, we extend Acc to include a recall-based adjustment for instances involving private libraries. This ensures that AccR not only evaluates the direct accuracy of code execution and question answering but also the inclusion and correct usage of private library functions. AccR can be computed as follows:

$$\text{AccR} = \frac{1}{N} \sum_{i=1}^{N} \mathbf{I}(C_i = \hat{C}_i) \cdot \mathbf{R}(C_i, \hat{C}_i), \tag{2}$$

$$\mathbf{R}(C_i, \hat{C}_i) = \frac{|\mathbf{F}(C_i) \cap \mathbf{F}(\hat{C}_i)|}{|\mathbf{F}(C_i)|}, \tag{3}$$

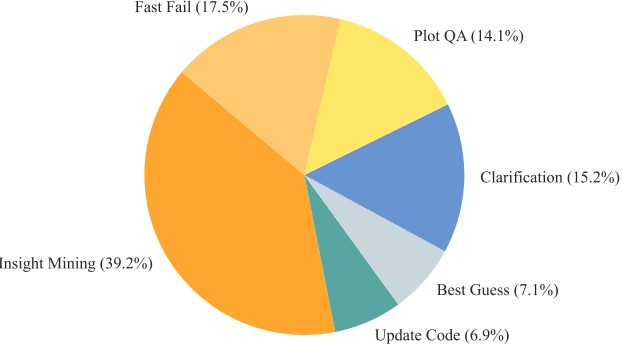

*Figure 3.* Distribution of ACTION Mode in CoTA. It contains 6 common types in conversational data analysis tasks.

where $\mathbf{R}(C_i, \hat{C}_i)$ quantifies the recall rate of relevant library functions in the predicted code. $\mathbf{F}(C_i)$ and $\mathbf{F}(\hat{C}_i)$ denote the set of private library functions in the reference codes and the set actually utilized by agents in the predicted codes, respectively. The final score would be weighted sum of Acc and AccR. We conduct an in-depth analysis of the impact of AccR on Private mode evaluation in Appendix M.

## 5. Dataset Quality Evaluation

To ensure the data quality of CoTA and the reliability of our proposed human-sandbox data generation, we conduct a comprehensive human evaluation focusing on both gen-

eral and action-specific aspects by selecting 500 samples randomly which is approximately 50% of the full dataset. To do this, we invite 10 experts with extensive data analysis experience **outside** authors to review the dataset. Details of evaluation instructions can be found in Appendix O.

**General Metrics.** Following (Hu et al., 2024), we conduct human evaluation on more NL metrics about reasonableness and coherence across turns of conversations. The results are shown in Table 3, where `w/o human` refers to the data which is fully annotated by LLM agents. And `w/ human` means the data in the COTA, which is annotated by human-sandbox collaboration. From the table, it shows, after human involvement, the acceptance ratio rises to approximately 0.95, indicating that the involvement of human annotators who are professionals in data analysis is sufficient to ensure the quality of the auto-generated dataset, thus demonstrating the balance of trade-off in DECSION COMPANY between efficiency and quality of complex data annotation workflow.

**Action-wise Metrics.** Human evaluation is also conducted with a focus on the actions. Figure 4 illustrates the consensus among experts that all actions in COTA are both necessary and commonly observed in real-world data analysis scenarios. All instructions and details can be found in Appendix O.

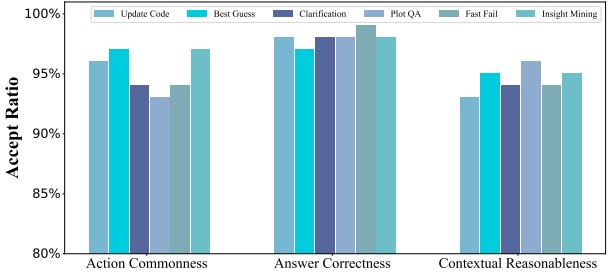

*Figure 4.* Results of human evaluation on action-wise metrics of the dataset quality.

# 6. Evolving LLMs Towards Conversational Data Analysis Agents

In this section, we discuss our approach of equipping LLMs as data analysis agents with tools and reasoning. We then introduce our self-generated reflection strategy, ACR, to enhance their performance in conversational settings.

## 6.1. Toolkit

Our tool sets include an executor, a user simulator, and a chart-to-table converter. The executor provides an environment for models to observe real-time feedback on their intermediate code results (Xie et al., 2023; Wang et al., 2024b). The user simulator (Wang et al., 2024b; Yan et al., 2023),

powered by GPT-4-Turbo, tests the models' ability to generate codes after clarifying details when facing under-specific questions. The chart-to-table (Liu et al., 2023a) converter mitigates the prevalent issue of LLMs' inability to comprehend plots by converting them into tables. Descriptions of these tool sets can be found in Appendix H.1.

## 6.2. Reasoning

Reasoning is a critical process for transitioning LLMs into data analysis agents (Huang & Chang, 2023). In COTA, we incorporate two primary reasoning methods for code generation and multiple-choice answers. The first is the **Chain-of-Thought (COT)** prompting technique (Wei et al., 2022), which enhances the complex reasoning abilities of LLMs by dividing the reasoning path into multiple steps. The second method is **CodeAct**, which enables models to make decisions by generating reasoning traces and codes in an interleaved manner (Yao et al., 2023; Wang et al., 2024a).

## 6.3. Adaptive Conversation Reflection (ACR)

Successful conversations are important since they encapsulate the logic necessary to meet user requirements and ensure correct steps of analysis or code generation. Motivated by this, we propose the **A**daptive **C**onversation **R**eflection (**ACR**) approach to enable data analysis agents to learn from successful user-code histories through a two-step process.

**Pseudo Code Logic Generation.** First, given the last previous history $(\mathbf{u}_{t-1}; \mathbf{a}_{t-1})$, when $t > 1$, we prompt the data analysis agent to reflect and generate its underlying logic $\mathbf{m}_{t-1} = f_\theta(\mathbf{u}_{t-1}; \mathbf{a}_{t-1})$, where $f_\theta$ refers to agent based on LLMs with parameter $\theta$. Also, $(x; y)$ represents two elements $x$ and $y$ are concatenated in the prompt. In our work, we consider the pseudocode to be $\mathbf{m}$, as it serves as an intermediate logic between natural language queries and codes.

**Re-Org One-Shot Reasoning.** Second, we re-organize them into a self-generated one-shot example with the order: $\mathbf{p}_{t-1} = (\mathbf{u}_{t-1}; \{\mathbf{m}_{t-1}; \mathbf{a}_{t-1}\})$, which represents the scenario where the input $\mathbf{u}_{t-1}$ is given, the agent should generate a logic $\mathbf{m}_{t-1}$ first, then generate answers $\mathbf{a}_{t-1}$. Finally, the data analysis agent can learn from $\mathbf{p}_{t-1}$ to first generate logic $\mathbf{m}_t = f_\theta(\mathbf{p}_{t-1}; \mathbf{u}_t)$ and generate an answer $\mathbf{a}_t = f_\theta(\mathbf{u}_t; \mathbf{m}_t)$ in the current turn $t$. When $t = 1$, we keep the same reasoning method of the original agent. Appendix G provides a detailed example for further illustration.

# 7. Experiments

## 7.1. Setup

**Models.** Our experiments primarily involve popular LLMs that are capable of generating code and following

*Table 4.* Overall results of LLMs in base, agent, and inter-agent modes on the CoTA dataset. **Pri-Act** refers to private library + action evaluation mode. **Agent** refers to agents using COT in code type and CodeAct in choice type. **Agent-R** involves textual reflection. **Multi-Agent** uses multiple LLM agents to collaborate. And **Inter-Agent** refers to our proposed method incorporating ACR.

| Model | Conversation Mode | | | | Answer Type | | Overall |
|---|---|---|---|---|---|---|---|
| | Normal | Action | Private | Pri-Act | Code | Choice | |
| Mistral-7B | 5.9 | 16.4 | 1.5 | 1.9 | 4.8 | 16.7 | 9.1 |
| Mistral-8 × 7B | 17.2 | 23.6 | 3.7 | 1.9 | 11.6 | 23.9 | 16.1 |
| CodeLlama-34B | 28.2 | 19.7 | 2.6 | 0 | 16.1 | 19.8 | 17.4 |
| GPT-4-Turbo | 30.8 | 18.5 | 6.4 | 3.7 | 18.9 | 18.5 | 18.8 |
| Claude-3-Opus | 22.6 | 28.7 | 2.6 | 5.6 | 14.5 | 29.0 | 19.8 |
| GPT-4-32k | 31.1 | 25.5 | 7.5 | 0 | 19.7 | 25.6 | 21.9 |
| Llama-3.3-70B | 33.6 | 32.4 | 6.7 | 3.7 | 21.4 | 32.7 | 25.5 |
| Claude-3.5-Sonnet | 37.9 | 36.3 | 10.5 | 9.3 | 25.6 | 36.6 | 29.6 |
| + Agent | 37.3 | 43.1 | 10.1 | 7.4 | 25.6 | 43.5 | 32.1 |
| + Agent-R | 36.2 | _48.5_ | 13.5 | 9.3 | 26.8 | _49.0_ | 34.9 |
| + Multi-Agent | **43.5** | 45.2 | _16.1_ | _11.1_ | _31.2_ | 45.5 | _36.4_ |
| + Inter-Agent | _42.9_ | **52.8** | **18.0** | **14.8** | **32.4** | **53.3** | **40.0** |

complex human instructions since this is a basic requirement in data analysis. Therefore, we investigate performance of 4 families of models, covering Mistral (Jiang et al., 2023), LLama (Roziere et al., 2023; Dubey et al., 2024), Claude, and GPT (Achiam et al., 2023) models.

**Implementation.** The implementations could be divided into 5 settings: 1) `Model-Base` refers to the LLM itself without reasoning and tool calls. 2) `Agent` mode involves multiple tool usage and reasoning. (COT + CodeAct) 3) `Agent-R` mode involves textual reflection for `Agent`. 4) `Multi-Agent` mode is implemented through the coordinated collaboration of multiple specialized agents, each responsible for a distinct set of functions inspired by (Liu et al., 2023d; Hong et al.). 3) `Inter-Agent` mode incorporates ACR as described in Section 6.3 beyond the AGENT. Further details can be found in Appendix I.

### 7.2. Experimental Results

Table 4 illustrates the comprehensive performance of popular LLMs and developed agent modes based on Claude-3.5-Sonnect on the CoTA. From the results, we can deduce the following: 1) CoTA is a challenging benchmark in which the SOTA method only achieves a score of 40.0%, leaving a large room for improvement. 2) Despite the performance of GPT-4-Turbo being nearly on par with GPT-4-32K in code generation, its overall performance still falls short of GPT-4-32K. This indicates that beyond code writing, understanding results, and analysis are equally important. Fortunately, the comprehensive settings of CoTA can assist users in selecting models for data analysis tasks. 3) We observe that CodeLlama frequently defines functions automatically

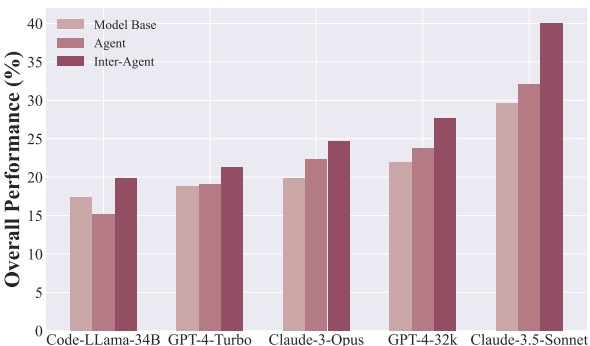

*Figure 5.* Visualization of the performance of Code-Llama-34B, GPT-4-Turbo, Claude-3-opus, GPT-4-32k, and Claude-3.5-Sonnet, all with base, agent, and inter_agent versions.

and applies these in the following code, thereby improving readability and logic. This is particularly beneficial in tasks related to data-analysis code generation. Such tasks often require the composition of API functions, which demands a profound understanding of the context and the ability to extract common patterns into reusable functions. By defining and reusing symbolic functions, CodeLlama can streamline complex contexts, making them more logical, which is an advantage for resolving complex tasks (Gu et al., 2023). 4) Claude-3.5-Sonnet and Llama-3.3-70B perform better than GPT-4 on base mode proving that our benchmark is not overfitting to the GPT family of models.

**LLM Agent Performance.** We also implement different agent modes for several mainstream LLMs. As shown in Figure 5, most models with `Agent` version outperform their

*Table 5.* Definitions and Examples of main error types.

| Error Type | Definition | Example |
|---|---|---|
| Key Error (23%) | Refers to the instance where the model imagines a reasonable but non-existent column name to retrieve the data table. | **Question:** *We want to find clients who have stable employment. We can consider stable employment as those with employment duration of 4 years or more.* 
 **Gold:** high_credit_long_duration[high_credit_long_duration[**'employment'**] == 'x>=4'] 
 **Error:** high_credit_long_duration[high_credit_long_duration[**'employment_duration'**] == 'x>=4'] |
| Lazy Assumption (39%) | Refers to the instance where the model tends to assume some middle results have already been prepared. | **Question:** *Now, I need to know the potential impact of the updated odds on the later rounds of the tournament.* 
 **Error:** # Assuming 'updated_odds_df' is already created and contains the updated odds 
 updated_odds_df = pickle.load('updated_odds_df.pkl') |
| Bad Instruction Following (49%) | Refers to the instance where the model can not follow instructions well, leading to the failure of answering questions. | **Question:** *Please answer the multi-choice question ... ...; you will need to generate Python code which can assist yourself to answer the question in this step.* 
 **Error:** 'D. None of above' 
 Explanation: There is no specific test or condition to check ... ... |

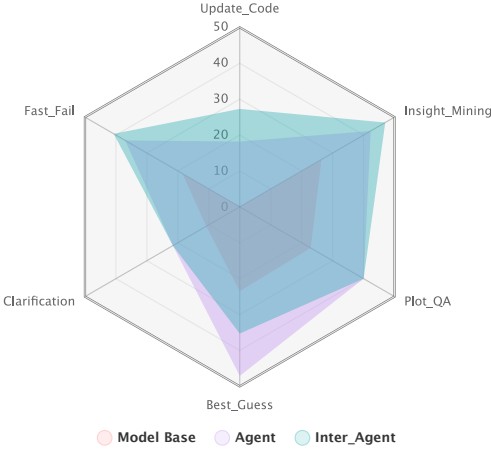

*Figure 6.* Visualization of the performance of GPT-4-32k across various categories in ACTION Mode. The comparison includes base, agent, and inter_agent versions.

base version, highlighting the crucial role of tools and reasoning in enhancing the performance of LLMs under complex tasks (Liu et al., 2024b; Xie et al., 2023). Also, all models exhibit obvious improvements in the `Inter-Agent` mode with ACR. This indicates that the underlying logic of successful conversation histories is instrumental in guiding LLMs to become more proficient data analysis agents in conversational settings.

**Fine-Grained Results on ACTION Modes.** Figure 6 provides a comparative evaluation of GPT-4 model across various ACTION modes detailed in Section 2. The conversational data analysis agent, `Inter-Agent`, obviously outperforms in most areas, especially in managing

`Fast_Fail` queries and executing `Update_Code` actions. However, it falls short in the `Best_Guess` action when compared to the `Agent`. We note that ACR tends to make agents overly tractable in re-org one-shot example $p_{t-1}$ and current generated logics $m_t$. If $p_{t-1}$ and $m_t$ do not contain instructions on making assumptions, agents tend to select `None of Above`. This observation suggests that excessive reliance on historical data may hinder the inherent ability of models to conjecture based on instant user behaviors. Therefore, striking a trade-off between user-code history exploration and real-time user conversation, especially in under-specific questions, is crucial for improving LLM agents' performance in conversational settings.

**Error Analysis.** We conducted an error analysis by sampling 200 error cases from each of 8 LLMs to gain insights into conversational data analysis as shown in Table 5. The errors were categorized into three main types: **(1) Key Error (23%)**, where the model incorrectly matches column names in the provided data table or references nonexistent values; **(2) Lazy Assumption (39%)**, where the model assumes that intermediate results or states are already available or saved on disk without verification; and **(3) Poor Instruction Following (49%)**, where the model fails to strictly follow instructions, resulting in incorrect answers. Furthermore, Figure 7 demonstrates that ACR effectively reduces errors in each category. The special common errors in private library mode are: (1) LLMs fail to retrieve useful packages, and usually retrieve excessive packages instead (67%); and (2) some LLMs show conservative usage of private libraries to minimize errors, which violate the user intent (39%). A more detailed error analysis can be found in Section N.

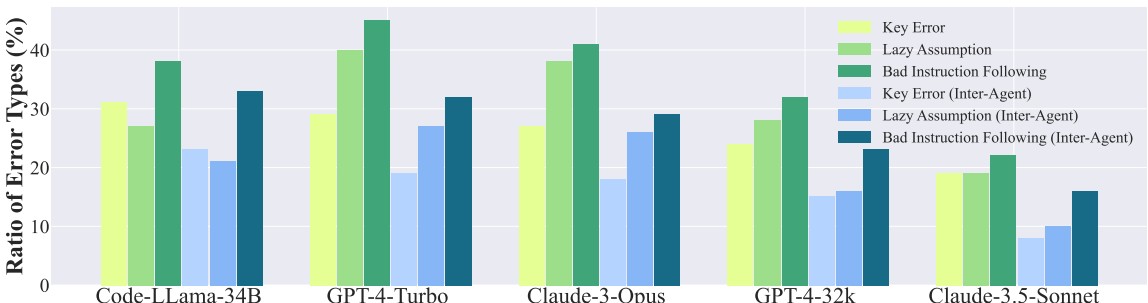

Figure 7. The visualization of different error types across different models and settings.

# 8. Related Work

**Large Language Models for Data Analysis.** The use of LLMs for data analysis has been a topic of interest in recent years. LLMs powered by In-Context Learning (Yang et al., 2023; Dai et al., 2023; Dong et al., 2024) have been employed in various data analysis tasks, such as SQL query generation (Pourreza & Rafiei, 2024; Gao et al., 2024; Lei et al., 2024; Zhang et al., 2024; Gu et al., 2024; Wang et al., 2025; Qu et al., 2024; Li et al., 2024c), pandas or python code generation (Jain et al., 2024; Chen et al., 2024; 2023a; Li et al., 2024b; Zha et al., 2023; Zhang et al., 2023a; Zheng et al., 2024b), and data visualization (Chen et al., 2023b; Huang et al., 2024a). However, most of these works focus on single-turn setting, where the user query is explicit and does not require any conversation or clarification. Recently, there has been a growing interest in conversational data analysis, where the user intents may need to be clarified or refined through conversations (De Vries et al., 2020; Yan et al., 2023; Wang et al., 2024b).

**Data Analysis Benchmarks.** The development of benchmarks for data analysis tasks has been a crucial factor in driving the progress of LLMs in data science. Existing benchmarks can be broadly categorized into single-turn and multi-turn benchmarks. Single-turn benchmarks, such as HumanEval (Chen et al., 2021), MBPP (Austin et al., 2021), Spider (Yu et al., 2018), BIRD (Li et al., 2023a), Text2Analysis (He et al., 2024), DABench (Hu et al., 2024) and DS-1000 (Lai et al., 2023), focus on generating code snippets or closed-form insight summaries for data analysis given a single user query. To explore conversational nature of real-world data analysis scenarios, where the user intent may need to be clarified or refined through conversational communication, several multi-turn benchmarks have been proposed, including CoSQL (Yu et al., 2019a), and ARCADE (Yin et al., 2023). However, these benchmarks are primarily focused on code generation and do not cover other aspects of data analysis, such as data visualization and understanding based on intermediate results. Our work extends the existing literature by introducing a new benchmark for evaluating LLM agents in conversational data analysis.

**Multi-Agent Environments for Data Generation.** LLMs have proven to be effective in constructing multi-agent environments for automatic data generation. For instance, Lu et al. (2023) and Ding et al. (2023) simulate dialogs for QA and text generation tasks. Also Li et al. (2023b) generates data about API calls using multi-agent environments. This is because LLM agents can simulate believable human actions when placed in an environment with dynamically updating knowledge and memory (Park et al., 2023). Inspired by this, we created DECISION COMPANY to generate conversation log data for data analysis with more behaviors. Unlike prior dataset-generation studies, we pioneer a benchmark for assessing conversational data-analysis agents.

# 9. Annotation Cost

Similar to BIGCODEBENCH (Zhuo et al., 2025) and SPIDER 2.0 (Lei et al., 2025), our annotators are recognized via authorship rather than direct payment. Working exclusively in a privacy-preserving, controlled interface, they logged a total of 3,677 minutes, incurring $71.39 in LLM usage, and annotator cost of $1,392.66 with $0.37875 per minute for professional data-analysis experts (Kazemitabaar et al., 2024; Liu et al., 2023b), yielding an overall benchmark development cost of $1,464.03, or roughly $1.45 per COTA instance, which is over four times cheaper than BIRD-SQL (Li et al., 2023a) ($6.13) and TABLEBENCH (Wu et al., 2024) ($6.00) despite COTA's higher complexity.

# 10. Conclusion

We introduce COTA, a new benchmark for evaluating LLM agents in conversational data analysis tasks. COTA is constructed via a scalable and expert-recognized high-quality multi-agent environment, DECISION COMPANY, and covers a wide range of practical scenarios. We evaluate data analysis agents based on popular LLMs on COTA, highlighting the challenges of conversational tabular data analysis. We also propose ACR, an effective reflection strategy for conversational data analysis agent evolution. Our experiments demonstrate that ACR can significantly enhance the performance of LLM agents, paving the way for future research.

## Acknowledgement

Reynold Cheng, Jinyang Li, Nan Huo, and Ge Qu are supported by the Hong Kong Jockey Club Charities Trust (Project 260920140), the University of Hong Kong (Project 2409100399), the HKU Outstanding Research Student Supervisor Award 2022-23, and the HKU Faculty Exchange Award 2024 (Faculty of Engineering). Bowen Qin was supported by National Science and Technology Major Project (Project 2022ZD0116306). Chenhao Ma was partially supported by NSFC under Grant 62302421, Basic and Applied Basic Research Fund in Guangdong Province under Grant 2023A1515011280, 2025A1515010439, Ant Group through CCF-Ant Research Fund, Shenzhen Research Institute of Big Data under grant SIF20240004, and the Guangdong Provincial Key Laboratory of Big Data Computing, The Chinese University of Hong Kong, Shenzhen.

## Impact Statement

This paper presents work whose goal is to advance the field of Machine Learning. There are many potential societal consequences of our work, none which we feel must be specifically highlighted here.

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

*Table 6.* Comparison of CoTA to popular LLM Agent-related datasets. CoTA represents a challenging dataset in data analysis with more comprehensive settings.

| | # Q | eval public available | multi-modal | private lib | multi-turn | conversation trajectory | data creation | output type |
|---|---|---|---|---|---|---|---|---|
| GAIA (Mialon et al., 2023) | 466 | ✗ | ✓ | ✗ | ✗ | ✗ | from-scratch | Text |
| ResearchAgent (Huang et al., 2023) | 15 | ✓ | ✓ | ✗ | ✗ | ✗ | semi | Multi-Types |
| SWE-bench (Jimenez et al., 2023) | 2290 | ✓ | ✗ | ✗ | ✗ | ✗ | from-scratch | Patch |
| AgentBench (Liu et al., 2024b) | 1091 | ✓ | ✓ | ✗ | ✓ | ✗ | semi | Multi-Types |
| RepoBench (Liu et al., 2023c) | 1669 | ✓ | ✗ | ✗ | ✗ | ✗ | semi | Code |
| DebugBench (Tian et al., 2024) | 4253 | ✓ | ✗ | ✗ | ✗ | ✗ | from-scratch | Code |
| **CoTA** | 1013 | ✓ | ✓ | ✓ | ✓ | ✓ | from-scratch | Code/Choice |

## A. Background Knowledge

**Requirements of client.** It naturally refers to specific data analysis tasks or questions that users want to accomplish, expressed in natural language.

**Result Type.** It typically refers to the format or nature of the output produced from analyzing data. Common result types include: "dataframes, lists, or various plot types."

**User intent.** It represents the underlying analytical goal or purpose behind a user's query.

## B. Data Resource

### B.1. CoTA

Our CoTA data is available under the lisense CC BY-SA 4.0.[1]

### B.2. Kaggle Tabular Data

The tabular data that we used to create CoTA are following open-source licenses: 1) **Public Domain**: Public Domain Mark 2) **CC-BY**: Creative Commons Attribution 4.0 International.

### B.3. Data Distribution

The Figure 8(a) visualizes our covered topics and domains.

## C. Comparison with Agent-related Benchmarks

Besides the benchmark comparisons in Table 2, we list more popular LLM Agent-related benchmarks comparisons in Table 6. Each column means: (1) **#Q**: This column represents the unique identifier or number assigned to each benchmark or dataset. (2) **eval public available**: This column specifies whether the evaluation metrics of the benchmark or dataset is publicly available for use. (3) **multi-modal**: This column shows whether the benchmark supports multi-modal data, meaning it can handle multiple types of input data (e.g., text, images) simultaneously. (4) **private lib**: This column indicates whether the benchmark or dataset includes a private library. (5) **Multi-Turn**: This column specifies whether the benchmark supports multi-turn conversations, which are conversations that involve multiple exchanges or steps. (6) **conversation trajectory**: This column indicates whether the benchmark involves conversation trajectories, which track the sequence and flow of conversations over time. (7) **data creation**: This column describes the method of data creation for the benchmark. "From-scratch" means the data was created anew specifically for the benchmark, while "semi" indicates that the data was created using a mix of new and existing data. (8) **output type**: This column specifies the type of output produced by the benchmark or dataset. Examples include "Text," "Multi-Types," "Patch," "Code," and "Code/Choice," indicating the nature of the outputs generated during evaluations.

---

[1] https://creativecommons.org/licenses/by-sa/4.0/deed.en

# D. Model Descriptions

In this section, we provide an overview of the various models used in our research. These models include both widely recognized and state-of-the-art LLMs that have been instrumental in advancing NLP tasks.

(1) **Mistral-7B-instruct-v01**: Mistral-7B is a powerful LLM designed to handle diverse NLP tasks. It is known for its efficiency in terms of parameter size while maintaining high performance. The 7 billion parameters enable it to process and generate human-like text effectively.

(2) **Claude-3.5-Sonnet**: Claude-3.5-Sonnet is an advanced version of the Claude series of LLMs. This iteration brings improvements in both accuracy and processing speed, making it a suitable choice for complex language understanding and generation tasks.

(3) **Mistral-8 × 7B-instruct-v01**: Mistral-8 × 7B represents a collection of 8 models, each with 7 billion parameters. This ensemble approach allows for enhanced performance through model averaging and provides robustness in generating more accurate results across different tasks.

(4) **CodeLlama-34B-Instruct-hf**: CodeLlama-34B is a specialized model focused on code-related tasks. With 34 billion parameters, it excels in code generation, code completion, and understanding programming languages, making it a valuable tool for software development and code-related research.

(5) **GPT-4-Turbo (gpt-4-0125-preview)**: GPT-4-Turbo is a highly optimized version of the GPT-4 model. It offers faster inference times and improved efficiency while maintaining the high-quality output that GPT-4 is known for. This model is particularly useful for applications requiring quick responses without compromising on quality.

(6) **Claude-3-Opus**: Claude-3-Opus is the latest in the Claude series, bringing substantial improvements in language understanding and generation. It integrates advanced techniques to enhance its contextual comprehension and generation capabilities, making it a top choice for sophisticated NLP tasks.

(7) **Llama-3.3-70B-instruct**: Llama-3.3-70B is a LLM with 70 billion parameters. This model is designed to tackle the most challenging NLP tasks, providing unparalleled performance in terms of accuracy and coherence in text generation.

(8) **GPT-4-32k**: GPT-4-32k is a variant of the GPT-4 model with an extended context window of 32,000 tokens. This extended context window allows it to handle long-form content more effectively, making it ideal for applications requiring extensive context retention and understanding.

## D.1. Model Shortcoming Analysis

**Long-Context Challenges.** The challenge of handling long-contexts is considerable in CoTA, especially for models with shorter maximum input lengths. Models such as Codellama-34B, which has a maximum input length of 16k, are particularly affected. For example, it is essential for LLMs to access all private function descriptions and codes for effective code generation with retrieved functions. The statistics shows that the average number of prompt tokens for PRIVATE is 15.7k, and notably, 20.9% of their prompts surpass the 16k length.

**Instruction Following.** Our experiments reveal that Claude-3.5-Sonnet requires minimal effort in prompt design due to its exceptional ability to follow human instructions. To be specific, only 3.2% of their results deviate from the provided instructions. However, other models exhibit a higher proportion of unexpected result types. For instance, extracting generated codes or answers from an LLM proves to be extremely challenging since it often embeds the answer in the middle of outputs rather than at the end as defined. We also observe that GPT-4-Turbo tends to generate longer codes in any setting. While this characteristic enhances its performance in code generation, it also results in 58.6% of the code generated during CodeAct reasoning being non-executable, thereby leading to incorrect answers. Furthermore, CodeLlama-34B-Instruct exhibits a lack of robustness when faced with longer or more complex prompts. With the addition of COT, the performance of CodeLlama significantly drops from 26.5% with simpler instructions to 19.5% in NORMAL code generation.

# E. Dynamic History Combination

## E.1. History Relational Database (H-RDB)

From all the User-AI conversation data shown in Figure 8(a), we split the User-AI conversation into several single-turn user queries and AI answers stored in a relational database, indexed by the conversational order as shown in figure 8(b). This

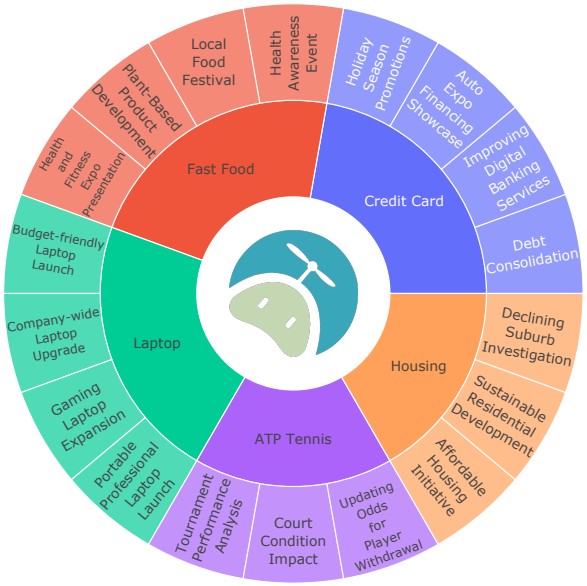

(a) Visualization of 18 topics and 5 data sources of CoTA.

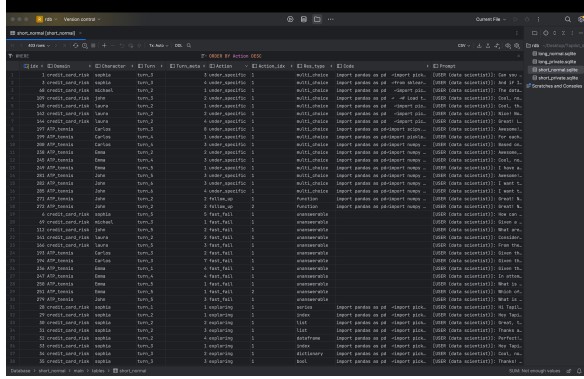

(b) The screenshot of History Relational Database (H-RDB).

*Figure 8.* Visualization of 18 topics and 5 data sources and constructed H-RDB of CoTA.

storage is subject to dynamic combinations for different scenarios.

### E.2. History Retrieval Queries

When retrieving the stored history information, we use `sqlite3`[2] python package. The search query is provided in sqlite3 format, for example: `SELECT {Prompt} FROM {table} WHERE 1=1 AND Domain = ?  AND ...`

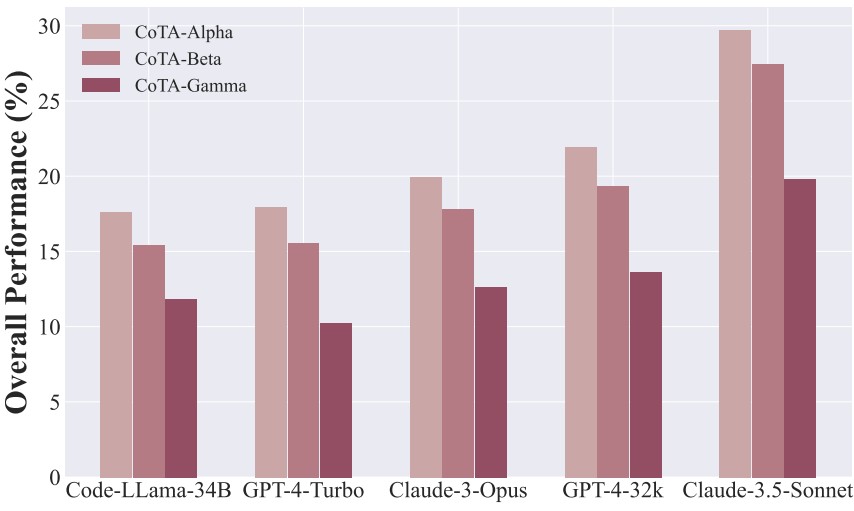

*Figure 9.* Visualization of the performance of CoTA-Alpha, Beta and Gamma.

---

[2]https://docs.python.org/3/library/sqlite3.html

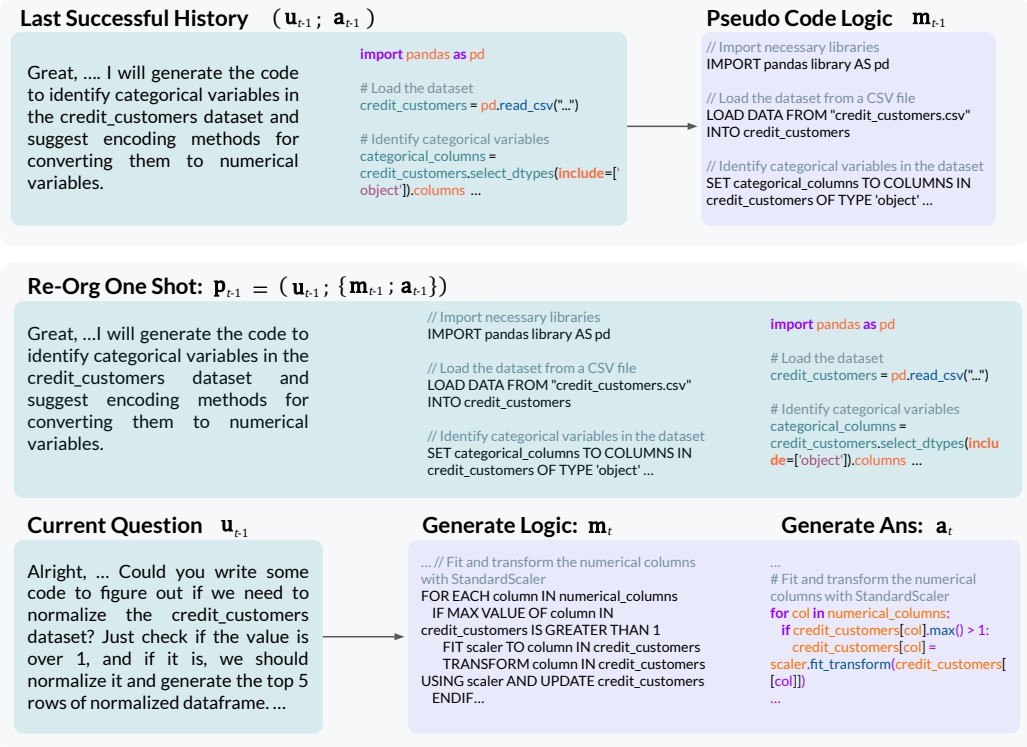

*Figure 10.* This is an overview of our proposed method, ACR. The areas highlighted in purple represent results generated by the agents.

## F. Dialog Types

CoTA can be categorized into **Statement-** (longer) and **Colloquial-** (shorter) dialogs. The statement-dialogs are more formal, resulting in more complex user instructions and code generations, which are commonly found in computational notebooks (Yin et al., 2023). On the other hand, colloquial dialogs involve shorter and simpler user questions, but exhibit more colloquial and conversational characteristics. This category of dialogs is primarily constructed through the process of prompting GPT-4 to segment and reinterpret the existing statement-dialogs.

## G. ACR Implementation

Figure 10 presents the detailed steps of ACR.

## H. AGENT Implementation

### H.1. Toolkit

**Executor.** To get the execution results of code generated by LLMs, we adopt Python Executor `exec()` which is implemented in Python [3], within a isolated Python environment. The output of the code execution, whether it be any return values, print statements, or error messages, is then captured by the Executor. This output is subsequently returned to the LMs, providing them with feedback on the results of their code generation to make a better next-step action or decision.

**User Simulator.** In addressing the clarification action type, LLMs are permitted to request clarification when they feel ambiguous about conditions from user queires. Therefore, we employ GPT-4 Turbo (with fixed version) to emulate the question-answering behavior of users, considering that GPT-4 has been demonstrated to provide feedback of equivalent quality to human responses (Wang et al., 2024b).

---

[3]https://docs.python.org/3/library/functions.html#exec

**Chart-to-Table.** We employ deplot (Liu et al., 2023a) to convert images into a table. Given the table, then LLMs can reason and answer the questions.

### H.2. Reasoning

**COT.** To evaluate the pure code generalization capability of data analysis, we restrict LLMs from executing code during generation. Therefore, we employ a zero-shot COT for the reasoning of the `Agent` mode. The key prompt to implement such COT is:

```
...  write a step-by-step outline and then write the code:
```

**CodeAct.** To evaluate analytical capabilities beyond mere code generation, we employ CodeAct for multiple-choice questions. Specifically, we set the `MAX STEP` for CodeAct reasoning to 5, with the Executor serving as the primary tool. Data analysis agents are tasked to generate, analyze, and draw conclusions about their results. If the result contains bugs, the corresponding message is returned to the agent for rectification, although this process may consume additional reasoning steps. We also manually provide a one-shot example to guide agents on how to CodeAct in CoTA. To prevent data leakage, we cross-reference examples across different tabular data. For instance, an example curated from `ATP_Tennis` could be used to guide LLMs in the `Laptop Pricing` dataset.

### H.3. Multi-Agent Implementation.

As a valuable and interesting agent type, Multi-Agent is recognized to have the potential to enhance performance in many reasoning tasks including BOLAA (Liu et al., 2023d) and MetaGPT (Hong et al.). However, MetaGPT was designed specifically for software development requirements without mechanisms for handling structured data and conversation histories, making it less applicable to our problem setting. Therefore we implemented the Multi-Agent reasoning type as introduced in (Liu et al., 2023d), which is a more general framework and can be implemented more flexibly in different settings. To be specific, except central CONTROLLER, we also create TOOL AGENT, CODE AGENT, DECISION-MAKING AGENT, and PRIVATE-LIB AGENT. Our results in Table 4 clearly indicate that the Multi-Agent configuration obviously outperforms the original Agent setting and model base setting, underscoring its potential. Notably, it achieves performance on par with our Inter-Agent configuration, particularly showing improvement in Multi-Choice tasks due to the important role played by the DECISION-MAKING AGENT. This supports one of our motivations: beyond code generation, providing insightful analysis for users based on results is crucial in data analysis tasks. However, we note that the Multi-Agent requires higher costs in our dataset and more sophisticated prompt design for each agent. Moreover, its performance begins to decay with increasing turns since each agent must be provided with not only conversations with users but also conversations between agents. This results in a prompt token consumption that is approximately 5.3 times higher than that of the Inter-Agent. Therefore, this observation reinforces the necessity of our design of ACR in the Inter-Agent. It is not only effective but also efficient, so it's more suitable in conversational settings.

## I. Implementation Details

### I.1. General Implementation

The `temperature` parameter is set to 0.0 for Claude, GPT-4, and GPT-4-Turbo and `top_p` to 1.0.

## J. Action Description

In this section, we categorize and formalize the action types in CoTA, identifying five distinct sub-categories that correspond to different types of user queries.

### J.1. Update_Code

The `Update_Code` action refers to instances where the user requests corrections for bugs or refinements to the conditions of previous queries.

### J.2. Fast_Fail

`Fast_Fail` is an action that alerts users when the current data contents or resources are insufficient to meet their requests, or when user queries contain factual errors.

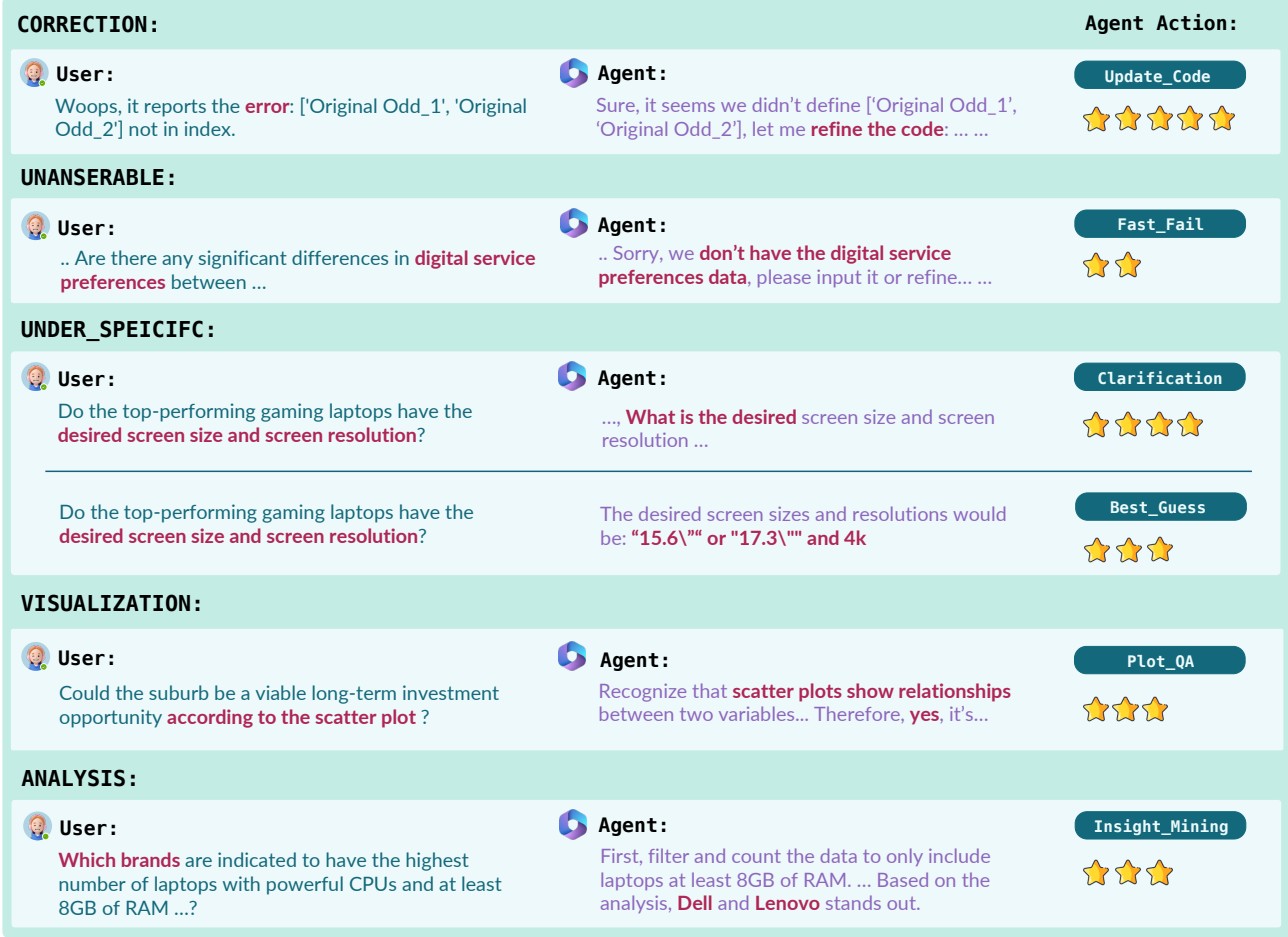

*Figure 11.* This figure provides an overview of action types in CoTA, illustrated by examples. We emphasize the keywords specific to each category, and demonstrate the relevant sections of the associated queries, as well as the agent actions. The number of ⭐ symbols represents the relative difficulty of each action. Please note that all free-text examples presented in this figure are only used for illustration purpose. **In CoTA, each answer format is limited to either code generation or multiple-choice questions.**

### J.3. Clarification

`Clarification` is a common action in response to under-specified questions, which are frequent in data-analysis queries. In this action, agents make the conditions of the question more specific and clear by seeking additional information from users.

### J.4. Best_Guess

While `Clarification` is an effective action to reduce the uncertainty, it can lead to issues such as user impatience due to unsteadily asking, and long dialog histories that result in attention distraction and long-context problems. Therefore, the `Best_Guess` action can address these issues by making appropriate assumptions based on data contents, domain knowledge, and commonsense knowledge for under-specific questions. However, there is also a risk that incorrect guesses can lead to hallucinations.

### J.5. Plot_QA

In real data analysis settings, agents are also expected to answer user questions about insights derived from plots. The `Plot_QA` action can assist users in better understanding the contents of plots for decision making.

## J.6. Insight_Mining

Beyond generating codes for users to retrieve expected results, code agents are also tasked with summarizing executed results from the environment to assist users in making informed decisions. This process, known as `Insight_Mining`, plays an important role in data analysis since it contributes to the evolution of code agents into comprehensive data analysis agents.

# K. Evaluation Metric Details

We introduce evaluation metric details in Section K.1, and implementations for each result type. And the distribution of result typs is presented in Figure 12(b)

## K.1. Evaluation Metrics

**Accuracy (Acc).** Acc is a common metric that evaluates ability of agents to generate code that executes correctly or to accurately answer multi-choice questions. It is defined as the proportion of instances where the predicted outputs match the expected reference output, across all evaluated tasks. For a given dataset with $N$ instances, where $C_i$ is the expected outcome (either execution result or correct answer) and $\hat{C}_i$ is the predicted outputs for the $i^{th}$ instance, Acc is calculated as follows:

$$\text{Acc} = \frac{1}{N} \sum_{i=1}^{N} \mathbf{I}(C_i = \hat{C}_i), \tag{4}$$

where $\mathbf{I}$ is an indicator function that returns $1$ if $C_i = \hat{C}_i$, and $0$ otherwise.

**Acc with Private Lib Recall (AccR).** Recognizing the importance of accurately leveraging specific user-defined libraries in code generation, we extend Acc to include a recall-based adjustment for instances involving private libraries. This ensures that AccR not only evaluates the direct accuracy of code execution and question answering but also evaluates the inclusion and correct usage of private library functions. AccR can be computed as follows:

$$\text{AccR} = \frac{1}{N} \sum_{i=1}^{N} \mathbf{I}(C_i = \hat{C}_i) \cdot \mathbf{R}(C_i, \hat{C}_i), \tag{5}$$

$$\mathbf{R}(Ci, \hat{C}i) = \frac{|\mathbf{F}(Ci) \cap \mathbf{F}(\hat{C}i)|}{|\mathbf{F}(Ci)|}, \tag{6}$$

where $\mathbf{R}(C_i, \hat{C}_i)$ quantifies the recall rate of relevant library functions in the predicted code. $\mathbf{F}(C_i)$ and $\mathbf{F}(\hat{C}_i)$ denote the set of private library functions in the reference codes and the set actually utilized by agents in the predicted codes, respectively. The final score would be weighted sum of Acc and AccR.

## K.2. DataFrame Comparison

The function compares two dataframes (`df_1` and `df_2`) by checking their indices, column presence, and column data. It uses `np.allclose()` for numeric data and direct comparison for non-numeric data. If a column in `df_1` is absent in the original dataframe, it searches for a matching column in `df_2`. The function returns `True` if `df_1` and `df_2` are equivalent, otherwise `False`. Please note, the `column names` will not be computed since different LLMs may have their only preference names. For example, the `win_ratio` generated by GPT-4 could be called `winning ratio` by Claude.

## K.3. Visualization Comparison

We note that it is hard to compare the closed-form results for visualization-based code generation since parameters of plots may be varied significant across models. For instance, GPT-4 generated plots may be the same with CodeLlama while their title names may be different, which leads to false negatives. Therefore we utilize `PIL` package to compute similarity between plots. To be specific, the function `compare_plots` takes two image file paths as inputs (`ai_output` and `reference_output`), resizes them to 800x600 pixels using the `LANCZOS` method, and saves them. The images are then read in grayscale mode to avoid the difference brought by colors. The function computes and returns the Structural Similarity Index (SSIM), a measure of image similarity, between the two images. This function can be used to compare an

AI model's output with a reference output. Finally, the code generated will be considered as correct if the similarity is larger than 0.6.

### K.4. Multi-Intent Evaluation

In this work, we evaluate the code generation performance on intent manner, which means if one user query contains multiple intents, then the total scores of this query will be the number of intents. We evaluate each intent separately and sum up the scores of all intents as the denominator when calculate the performance of each model in percentage.

### K.5. Private Function Recall

We notice that some LLMs tend to import as many as possible private functions while not using all of them. Thus, to extract all indeed used private functions in the customized function library, we utilize `AST` package. After extracting the used private functions, we calculate the recall coefficient according to Equation 6.

### K.6. Code Similarity Equivalance (CSE)

In the context of CoTA, the complexity of code generation tasks—many of which yield a score of zero—presents huge challenges in evaluating performance through Acc or AccR only. This is particularly evident when distinguishing between codes that differ by merely a single line of error or output, both of which would result in an Acc or AccR of zero, despite their obvious differences in code generation capabilities. To overcome this limitation, we propose the introduction of Code Similarity Equivalence (CSE), a nuanced evaluation metric designed to assess the similarity between generated codes and reference codes. Given that these codes originate based on identical user instructions, a high degree of similarity is expected. Our approach leverages a hybrid combination of models to reduce the bias, incorporating CodeT5+ and OpenAI Ada (`text-embedding-ada-002`) models, which are affordable and available for most institutes. This combination has demonstrated a strong correlation with human evaluative preferences, offering a more refined and accurate measure of code generation performance.

**Details.** We introduce here about how to conduct more nuanced evaluation of Acc or AccR with CSE. 1) We collect 180 instances of code generation including both NORMAL and PRIVATE. To evaluate the quality of these codes, we enlist experts who are proficient in data science and Python as evaluation committee.
2) They evaluate code generated by several models, including GPT-4-32k, GPT-4-Turbo, CodeLlama-Instruct-34B, Claude-3.5-Sonnet. Each evaluator is provided with comprehensive user code histories, tabular contents, the current query, access to the `decision_company` private library. Please note that evaluations are conducted only based on their expertise and experience, without any predefined guidelines and discussion, to avoid bias.
3) We ask for a relative ranking of generated codes among models over absolute scoring to avoid potential variability in scoring preferences among the evaluators.
4) In cases of parts of divergent rankings, the evaluators engage in discussions regarding the specific code samples until a consensus was reached. This step ensures a more reliable and agreed-upon evaluation outcome.
5) The evaluation committee then examine various open-source and readily available embedding models to measure code similarity, aiming to closely match their ranking preferences. Our exploration identifies that the score system consisting of CodeT5+ (Wang et al., 2023) and Ada (`text-embedding-ada-002`) most closely aligned with human evaluative preferences.

**Introduction of a Mixed Evaluation Metric (AccSE & AccSER).** To accurately reflect the nuanced capabilities of code generation models, we propose a composite metric that integrates Code Similarity Evaluation (CSE) with Accuracy (Acc), termed Accuracy for Similarity Evaluation (AccSE). This metric is concisely defined as:

$$\text{AccSE} = \begin{cases} 1.0, & \text{if } C = \hat{C}, \\ 0.5, & \text{if } S_1 > 0.85 \wedge S_2 > 0.9, \\ 0.25, & \text{if } (S_1 > 0.85 \wedge S_2 \leq 0.9) \\ & \vee (S_1 \leq 0.85 \wedge S_2 > 0.9), \\ 0, & \text{otherwise.} \end{cases} \tag{7}$$

Where:

- $C$ and $\hat{C}$ represent the reference and generated code execution outcomes, respectively.

- $S_1$ denotes the CSE score based on CodeT5+.

- $S_2$ denotes the CSE score based on Ada.

This formulation succinctly captures the evaluation criteria for AccSE, with symbols $S_1$ and $S_2$ representing the CSE scores based on CodeT5+ and Ada, respectively. The logical operators $\wedge$ and $\vee$ are used for "and" and "or" conditions, respectively, to further compact the notation. AccSER is computed in the similar way just times recall score for each value as Eq. 6.

We hold this for future evaluation system of CoTA when we conduct more extensive cases with involved with more expert volunteers.

**Rationale Against GPT-4-Based and Multi-Agent Evaluation Methods.** While existing research suggests that GPT-4-based soft evaluation could enhance the assessment of complex generative tasks, such approaches are deemed unsuitable for CoTA due to several critical reasons:

1) **Bias Concerns:** The prototype annotations and questions in our study originate from a GPT-4-based agent environment. Employing GPT-4 for evaluation purposes could inadvertently introduce a self-enhancement bias (Zheng et al., 2024a), compromising fairness across model evaluations.

2) **Cost Concerns:** Although multi-agent evaluation frameworks, incorporating diverse families of LLMs, is to mitigate bias (Li et al., 2023c), the economical and computational overhead is obvious. Specifically, evaluations in such settings require at least twice the token consumption than that used in generation alone, rendering it impractically expensive in CoTA.

Given these considerations, our research proposes an alternative evaluation methodology that is both cost-effective and reliable for evaluating the accuracy of complex data science code generation at this time. We demonstrate that CodeT5+, a remarkably efficient code embedding model, can obviously distinguish between varying performance levels and accurately identify correct code logic. Crucially, this model offers a pragmatic balance between evaluation thoroughness and resource efficiency.

### K.7. Other Value Types

For other result types, such as dictionry, set, list, we directly compute the exectued results and determine whether they are equal or not.

### K.8. Case-by-Case Evaluation

While we categorize instances according to result types and provide evaluation codes for each type, some scenarios requires a case-by-case evaluation script. For instance, in most dataframe or matrix comparisons, we employ `np.close()` and `string` match for result comparison. However, in some cases, such as using a dataframe or matrix to display a classifier's Confusion Matrix, the predicted code is deemed correct if its `f1-score` surpasses that of the referenced code, even if their `f1-scores` are not similar. For the evaluation script of CoTA, we manually review and adjust the scripts to accommodate each case.

### K.9. Evaluation Script Caching

Each example will be provided by a specific evaluation script to ensure the precision of our assessments (Lai et al., 2023). To manage the extensive effort required to design scripts for each example, we introduce a cache-based evaluation binding approach. Initially, we classify mainstream result types, which are collected through steps introduced in Section 3.1, and develop highly specialized scripts for each type, such as dataframes and dictionaries. When new data is generated, an evaluation script is automatically assigned based on the result type. Annotators then review the assigned script to ensure its accuracy; if necessary, they adapt and generate a new script tailored to the specific case. This method allows us to streamline the evaluation process, making it more efficient with minimal human intervention. The details of the evaluation script in each result type can be found in Section K.

## L. ACTION Evaluation Mode

### L.1. Correction

**Update_Code.**   This could be evaluated within a static setting where the bug feedback is embedded into user-code history. Agents are requried to update the previous code via user feedback.

### L.2. Unawserable

**Fast_Fail.**   In DECISION COMPANY, we keep the original unanswerable questions and categorize them as multi-choice questions. This is done to evaluate if agents can identify these questions based on their analysis of table contents and commonsense knowledge. To prevent any biased setting, such as specially designed prompts that might mislead agents into determining a question as unanswerable, we sample an equal number of under-specified problems and answerable questions. We then reformulate their choices, enabling the model to decide whether a question is answerable with clarification or assumption, or to directly classify it as unanswerable.

### L.3. Under_Specific

**Clarification.**   To evaluate the performance of agents on clarification action, we employ a dynamic setting that incorporates a User Simulator. This simulator mimics user feedback based on the ground truth code or answer. Initially, conversational data analysis agents are expected to pose questions for clarification, simulator will answer it according to the ground truth answers. Subsequently, these agents are tasked with generating the final code, understanding both the original history and the history of clarifications. This setup provides a robust assessment of the agents' ability to converse with human, clarify ambiguities, and generate accurate code.

**Best_Guess.**   We aim to evaluate the ability of conversational data analysis agents to make accurate assumptions when faced with ambiguous questions, without resorting to constant clarification, which could potentially frustrate users. We believe that the best guess of an agent should not impact the final decision and this evaluation metric should be somehow flexible. For instance, in a credit card application scenario, the term `young people` could refer to individuals aged 20-40 or 25-45, making it challenging to be evaluated by fixed metrics. Therefore, we opt to use multiple-choice questions to assess the agents' assumption-making capabilities. We posit that an assumption is appropriate only if it does not influence the final decision-making process.

### L.4. Visualization

**Plot_QA.**   We evaluate the analysis capability of agents around plot in COTA. The end format of answer would be multiple choices.

### L.5. Analysis

**Insight_Mining.**   We evaluate the analysis capability of agents generally in COTA. We opt to use multi-choice questions to evaluate it.

## M. Private Mode Analysis

**Overall Results.**   Table 4 and Figure 12(a) indicates that the PRIVATE setting presents a considerable obstacle, with the best performing Claude-3.5-Sonnet `Inter-Agent` only achieving 18%. This demonstrates that understanding and implementing user-specific functions is a critical and urgent skill for LLM agents in real-world data analysis tasks (Zan et al., 2022).

**The Critical Role of Function Relative Recall.**   Notably, CodeLlama outperforms GPT-4-Turbo in Acc within the PRIVATE setting. However, its performance declines significantly relative to GPT-4-Turbo upon the consideration of private library relative recall in the generated codes, as measured by the AccR metric. This observation suggests that CodeLlama tends to reply less on user-defined private functions, aiming to reduce risk of code errors. Therefore, AccR metric can spotlight the balance required between proficient code generation and the meticulous integration of user-specified private libraries to foster safer and more satisfying code production.

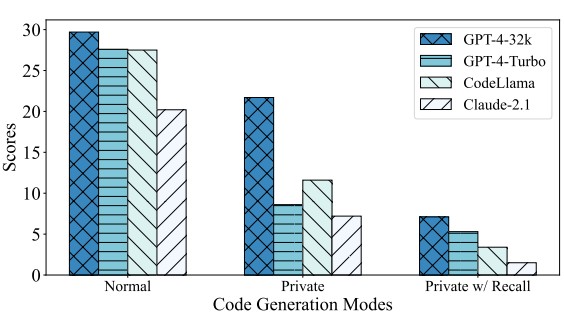
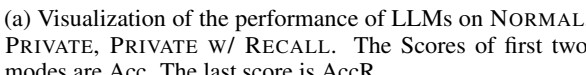

(a) Visualization of the performance of LLMs on NORMAL, PRIVATE, PRIVATE W/ RECALL. The Scores of first two modes are Acc. The last score is AccR.

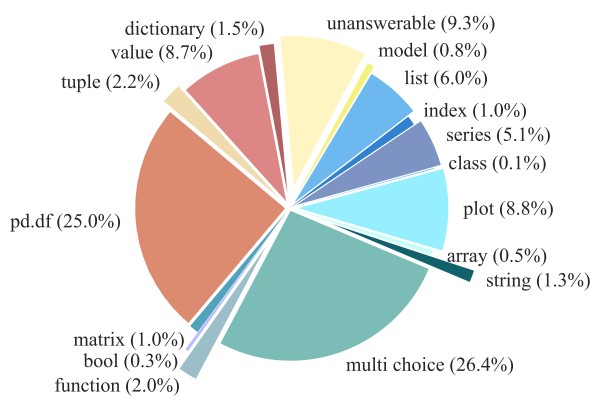

(b) Visualization of the results types distribution of CoTA.

*Figure 12.* Visualization of the performance of LLMs with PRIVATE mode and results types distribution.

## N. Detailed Error Analysis

In this study, the error patterns exhibited by tested LLMs are critically examined to provide insight into the predominant challenges faced during their operation. A detailed discussion is provided in Table 5. We analyze 200 randomly sampled instances, categorizing errors into three main types as follows: **(1) Key Error (23%)** occurs when the model incorrectly assumes the existence of a column name in the provided data table which does not exist. This error reflects a fundamental misinterpretation of the table information and hallucination where the model uses non-existent fields for data retrieval. An example is the model's incorrect use of '`high_credit_long_duration['employment_duration']`' instead of the correct attribute '`['employment']`'. This error type suggests that the model may overly rely on its trained patterns rather than accurately assessing the real structure of the data, leading to 'hallucinated' column references. **(2) Lazy Assumption (39%)** refers to instances where the model tends to assume that intermediate results or states are already available or saved on disk. This often leads to erroneous or incomplete code execution paths, such as the premature use of '`updated_odds_df`' without ensuring its prior creation and calculation. This type of error may arise because models often seek shortcuts in their processing, opting to retrieve and manipulate existing objects rather than generating solutions from scratch. This tendency can reduce the reliability and flexibility of the model, as it may fail under conditions where dependencies are not pre-established. **(3) Bad Instruction Following (49%)** describes the model's failure to adhere strictly to given instructions, resulting in an inability to properly answer the posed questions. This is exemplified by the response of the model with 'D. None of above' when asked to generate Python code to help solve a question, showing a lack of direct engagement with the query requirements. This type of error often becomes more pronounced in later conversation turns, suggesting a compounding of misunderstandings or a degradation of context over the course of a session. These error types are critical in understanding the limitations of the current model, guiding future improvements in LLM agents' evolving abilities. Understanding these patterns helps in pinpointing specific areas where training data, model architecture, or conversation protocols need enhancement to improve overall performance and reliability.

**Error Type Analysis Across Different LLMs and Conversation Settings.** Figure 7 showcases the distribution of error types across various LLMs and settings. This figure provides a comparative insight into the frequency of three primary error types: Key Error, Lazy Assumption, and Bad Instruction Following, both in Model-Base and Inter-Agent setting. **Key Error** rates vary significantly across models. For instance, Code-LLama-34B exhibits a notably higher rate of Key Errors compared to other models, which might suggest a less effective understanding or integration of database schema information in this model. **Lazy Assumption** errors are consistently high across all models, indicating a common model behavior where assumptions are made about the state of computations or data availability. This could reflect an inherent model optimization to minimize computational expense by reusing existing data states or structures, which, while efficient, can lead to inaccuracies when those states are not correctly initialized or updated. **Bad Instruction Following** shows a general high trend across models, particularly noticeable in settings involving Inter-Agent conversations. This suggests that as models engage in more complex dialogues or tasks requiring cooperative problem solving, their ability to follow detailed instructions without deviation diminishes. This could be due to accumulating contextual misunderstandings or the increasing complexity of managing multiple instruction streams. **Inter-Agent Variations** are particularly interesting; while Key Errors

and Lazy Assumptions increase slightly, Bad Instruction Following errors show a marked increase. This may be due to the added complexity of coordinating and maintaining consistent task strategies between agents, highlighting a critical area for further research and model training refinement. These insights are crucial for understanding specific weaknesses in current LLM implementations and point towards necessary areas for improvement in model training protocols and architecture adjustments. Enhanced training methods focusing on schema understanding and multi-agent coordination could mitigate some of these prevalent errors.

**Private Library Mode Common Errors.**   Given the private-lib based code generation such as (Zan et al., 2022; Li et al., 2023b) usually follows a Retrieval-Augmented Generation (RAG) workflow. The common types of errors can be divided into the following categories:

- **Retrieval Phase Errors:** LLM-based models prioritize high recall over precision, retrieving excessive packages. Even the best inter-agent performance only achieves a 0.73 F1 score.

- **Generation Phase Errors:** Error patterns vary significantly by models. CodeLlama shows conservative usage of private libraries to minimize errors. Instead, GPT-4-Turbo demonstrates higher private library utilization but increased error rates. We introduced this in Appendix N.

Also, the general error types depicted above in Section N also exist. Therefore, developing a more advanced RAG system would be helpful.

## O. Dataset Quality Evaluation Details

### O.1. Human Evaluation

To evaluate the quality of the dataset annotation, we also conduct a thorough human evaluation. We select 500 samples and invite 10 experts with extensive data analysis experience to review the dataset. The evaluation metrics listed below are carried out using a binary scoring system, with scores of 0 or 1 (Reject or Accept). We divide the measurement metrics into two levels: **General Metrics** and **Action-wise Metrics**, which are elaborated below.

**General Metrics.**   To ensure the overall quality of the dataset, we apply a set of comprehensive general metrics. These metrics are designed to evaluate the ability of dataset to capture meaningful, diverse, and coherent multi-turn conversations in the data analysis domain. Here is a brief introduction:

- **Conversation Coherence:** Evaluate whether the dataset contains logically consistent conversations including generated codes that flow naturally across multiple turns and lead to expected answers. If yes, score 1; if no, score 0.

- **Scenarios Diversity and Reasonableness** Assess whether the dataset contains a wide range of scenarios without duplication, tasks, and user intents. And whether they are reasonable and can be fully supported by the given tabular data. This metric is satisfied if both sub-metrics are satisfied. If yes, score 1; if no, score 0.

- **Conversation Topic Coherence:** Measure the overall relevance of the conversation to the given topic, ensuring that the conversation stays on track and go off the data analysis questions. This metric is satisfied if both sub-metrics are satisfied:

  - Conversation Goal Relevance: Evaluate whether each conversation turn contributes meaningfully to achieving the final goal. Turns must remain focused on the final conversation goal. If yes, score 1; if no, score 0.
  - Table Relevance: Measure whether at least 40% of the conversation turns involve conversation with the provided tabular data for specific conditions. This ensures the conversation is sufficiently relevant to the dataset being analyzed. If yes, score 1; if no, score 0.

- **Ethics and Bias Representation:** Assess whether the dataset avoids biased, harmful, or unethical content. If yes, score 1; if no, score 0.

- **Conversation Naturalness:** Measure whether the conversation in the dataset reflect natural, conversational language, avoiding overly robotic or AI-like responses. Conversations should resemble real human conversations in tone and flow. If yes, score 1; if no, score 0.

- **Evaluation Scripts Quality:** This metric is satisfied if all sub-metrics are satisfied.

  - Tool Reliability: Measure whether tools usage logs such as the logs of user simulator tool are reasonable and trustworthy, and whether the Python executor tool utilizes common, reliable packages to ensure consistent and accurate results. If yes, score 1; if no, score 0.
  - Evaluation Script Flexibility and Comprehensiveness:
    1. Flexibility: Ensure that the evaluation scripts are not overly rigid and can accept multiple valid, reasonable outputs as correct, allowing for variations in agent responses. If yes, score 1; if no, score 0.
    2. Comprehensiveness: Measure whether the scripts are robust enough to handle a wide range of scenarios, including corner cases, ensuring they effectively evaluate all possible outcomes. If yes, score 1; if no, score 0.
    3. For multi-choice questions, the provided options should be valuable and challenging enough, where can't be easily figured out from merely question and options. If yes, score 1; if no, score 0.

- **Evaluation Script Scalability:** Measure how easily the evaluation scripts can be extended or adapted to accommodate new data. This metric evaluates whether the framework allows for seamless integration of new evaluation scripts without requiring significant modifications, ensuring efficient scalability as the dataset grows. If yes, score 1; if no, score 0.

**Action-wise Metrics.** In addition to the general metrics, we apply specific metrics to evaluate the detailed actions captured in the dataset. These action types include `Update_Code`, `Fast_Fail`, `Clarification`, `Best_Guess`, `Plot_QA`, and `Insight_Mining`.

Each action type is evaluated using core metrics to ensure its relevance, accuracy, and contextual consistency within the dataset. These metrics guarantee that the dataset reflects realistic and valid scenarios in the data analysis domain. Here is an overview:

- **Action Commonness in Data Analysis:** Evaluate whether the action cases are commonly seen in real-world data analysis tasks. This ensures that the dataset captures authentic, practical scenarios that analysts frequently encounter. If yes, score 1; if no, score 0.

- **Correctness of Reference Answers:** Assess whether the ground-truth or reference answers provided in the dataset are accurate. The dataset should provide correct, verifiable solutions to the user query represented in each action. If yes, score 1; if no, score 0.

- **Contextual Reasonableness:** Measure whether the context surrounding the action is comprehensive and logical. This ensures that the action occurs within a reasonable, coherent dialogue flow, taking into account all relevant factors from previous turns. If yes, score 1; if no, score 0.

**Dataset Quality Statistics.** We compare the quality of the 500 sampled annotated data before and after human calibration, where the pre-calibration data was fully annotated by large language models (LLMs). As shown in Table 3, there is an obvious improvement in dataset quality following human calibration performed by annotators. The low acceptance ratio of the data prior to calibration underscores the necessity of this process. After calibration, the acceptance ratio rises to approximately 0.95, indicating that the involvement of annotators who are experienced in data analysis is sufficient to ensure the quality of the dataset, thus demonstrating the trade-off between efficiency and quality of our annotation workflow.

Specifically, the metrics for Scenario Diversity and Reasonableness improved from 0.46 to 0.96, reflecting enhanced coverage and variety of scenarios as shown in Section 3.1. Moreover, Conversation Topic Coherence increased from 0.17 to 0.93, indicating better alignment with the conversation topics. The Ethics and Bias Representation metric achieved a perfect score of 1.00, confirming the adherence of the dataset to ethical standards, while Conversation Naturalness improved from 0.67 to 0.95, suggesting more natural, human-like dialogues. Additionally, our Evaluation Scripts Quality and Evaluation Scripts Scalability scoring 0.98 and 0.94, respectively, prove an efficient solution to be against with data leakage problems. Overall, these results highlight the effectiveness of human calibration in enhancing both the conversation quality and the robustness of DESCISION COMPANY and high quality of CoTA.

Human evaluation is also conducted with a focus on the actions. Figure 4 illustrates the consensus that all actions in CoTA are both necessary and commonly observed in real-world data analysis scenarios. Furthermore, our simulated scenarios successfully capture and reflect key characteristics of these real-world conversational data analysis conversations.

*Table 7.* Package Diversity of Our Dataset

| Category | pandas | matplot | Machine Learning (sklearn, scipy, seaborn) | numpy |
|---|---|---|---|---|
| **Percentage (%)** | 56.47% | 7.90% | 16.78% | 12.64% |

## O.2. Dataset Diversity

We acknowledge the importance of data diversity and believe our benchmark, COTA, effectively demonstrates it across multiple dimensions. We measure data diversity through several aspects:

- Domain / Topic Diversity, shown in Figure 8(a).

- Result Type Diversity, covering a wide range of query types requiring different code techniques, as displayed in Figure 12(b).

- Action Diversity, presented in Figure 3, where we demonstrate the comprehensive coverage of various action types in conversational data analysis. To our knowledge, we are the first to offer such extensive action type diversity in this domain.

*Table 8.* Average Unique 3-grams

| Metric | CoTA (1013) | DS-1000 |
|---|---|---|
| Average unique 3-grams in reference_answer (including codes & multi-choice answers) | 46.64 | 11.63 |
| Average unique 3-grams in current_query | 84.54 | 115.31 |
| Average unique 3-grams in prompt_with_context (history) | 1033.40 | 115.31 |

*Table 9.* Total Unique 3-grams Counts

| Metric | CoTA (1013) | DS-1000 |
|---|---|---|
| Total unique 3-grams in reference_answer | 47,245 | 11,633 |
| Total unique 3-grams in current_query | 85,643 | 115,305 |
| Total unique 3-grams in prompt_with_context (history) | 1,046,832 | 115,305 |

Additionally, we evaluate data diversity through two more specific metrics:

- Package Diversity: Table 7 shows the distribution of query topics that cover various Python packages commonly used in data analysis, such as pandas, matplotlib, and machine learning libraries (sklearn, scipy, seaborn), with pandas dominating at 56.47%, followed by machine learning packages (16.78%) and numpy (12.64%).

- Query Diversity: Following the methodology in (Li et al., 2023a), we compute n-grams ($n = 3$) to reflect the diversity of each query. We compare COTA against DS-1000 (Lai et al., 2023), a popular data analysis benchmark, to show the diversity of queries in our dataset.

  - Average Unique 3-grams: Table 8 illustrates that COTA provides a much higher diversity in reference answers, averaging 46.64 unique 3-grams compared to 11.63 of DS-1000. For current queries, COTA maintains a comparable level of diversity, but when context is considered (prompt_with_context), COTA vastly outperforms DS-1000, demonstrating its ability to handle complex, context-driven conversations with an average of 1033.40 unique 3-grams.

- Total Unique 3-grams Count: It show the diversity of the whole dataset. Table 9 further supports the diversity of CoTA, with the total number of unique 3-grams in reference answers reaching 47,245, significantly surpassing 11,633 of DS-1000. The total unique 3-grams in current queries stand at 85,643, and in prompt_with_context, CoTA achieves an outstanding 1,046,832 unique 3-grams, compared to 115,305 of DS-1000. This further highlights the rich contextual conversations captured by CoTA.

Overall, these metrics, along with the diversity in domains, result types, and actions, underscore extensive coverage of real-world data analysis tasks in CoTA. It surpasses existing benchmarks in capturing the full range of complexity and diversity needed for robust evaluation of conversational data analysis.

## P. Ethical Statement

The application of LLMs for automatic data generation requires a rigorous examination of ethical implications. The primary concern is the potential for LLMs to generate contents that could be considered harmful or biased. To mitigate these risks, human annotators already filter and fix all problematic cases in Section 3.2. Also, LLMs may disseminate private or sensitive information. Therefore, we employ anonymization techniques wherein personal identifiers are systematically altered. For example, the name strings are replaced randomly, and any information of personas are switched as well. And the geographical locations of `John Smith` will be replaced with locations of `Carlos Garcia` to prevent any linkage to real-world individuals or entities. These procedures are conducted in Section 3. Moreover, we are committed to ensuring that the outputs generated by our LLM, referred to as CoTA, are free from political or sexual biases. To this end, each output, including conclusions and generated responses, is rigorously reviewed by the authors. In a nutshell, our ethical framework is built on a foundation of transparency, accountability, and a proactive stance towards mitigating any ethical concerns associated with the use of LLMs. The measures we have implemented reflect our commitment to upholding the highest standards of ethical research practice with LLMs

## Q. DECISION COMPANY Prompt

The process begins with the generation of client personas, as shown in Figure 13, where the Administrator agent is prompted to create meaningful personas. Following this, we simulate diverse analysis scenarios using In-Context Learning (ICL), which is depicted in Figure 14, allowing us to explore a wide range of potential outcomes. A critical aspect of the system is the discussion of analysis plans, where the conversation between the Data Scientist agent and the Client agent, illustrated in Figure 15, results in the generation of a series of analysis plans. To further support the process, conversation logs are annotated to capture the essence of conversations, with Figures 16 and 17 showing the perspectives of the Data Scientist Agent and the AI Chatbot Agent, respectively. Lastly, the evolution of our private library is detailed in Figure 18, which demonstrates the framework for prompting GPT-4 to generate code automatically, while human intervention plays a key role in minimizing bias and correcting errors.

## R. Implementation Prompt

### R.1. CODE GENERATION

The Figure 19 describes how we prompt LLM model to generate code to answer user queries. And Figure 20 describes how we prompt LLM in `Agent` to generate code to answer user queries following with chain-of-thought (Wei et al., 2022). Finally, Figure 21 describes how we prompt LLM in `Inter-Agent` to generate code to answer user queries with our proposed **ACR**. And Figure 25 describes how we prompt LLM in `Model-Base` version to ask for clarification.

### R.2. MULTI-CHOICE

The Figure 22 describes how we prompt LLM to answer user queries. And Figure 23 describes how we prompt LLM in `Agent` to answer user queries following with CodeAct (Wang et al., 2024a). Finally, Figure 24 describes how we prompt LLM w/ `Inter-Agent` to answer user queries with our proposed **ACR**.

**Find Target Group:**

In Decision Dynamics, a company specializing in data-driven decision-making support, CEO Patrick Reynold leads a team of experts dedicated to assisting clients and organizations in making informed choices for any domains based on comprehensive data analysis.

You are an administrator at Decision Dynamics, Sarah Johnson, responsible for receiving client inquiries and managing business budge to make sure profits. Your role involves collaborating with data scientists to calculate the costs of each case and determining whether to accept or decline orders. Additionally, you oversee the cost control of dialogue turns generated while resolving cases. Today, you need to find out target clients for tabular data about fastfood with the following columns: ['restaurant', 'item', 'calories', 'cal_fat', 'total_fat', 'sat_fat', 'trans_fat', 'cholesterol', 'sodium', 'total_carb', 'fiber', 'sugar', 'protein', 'vit_a', 'vit_c', 'calcium', 'salad'].

-- Please search out 3 groups of clients whose busniess needs this data. Fill them in the JSON file: {'Goal': [Relation to this tabular data], 'Group': [Group name]}. Each JSON should be split by ' || ', and no numbers are needed. Make sure 3 Group have 3 different business needs:

**Agent Output:**
{'Goal': 'Recommend restaurants based on users' nutritional needs', 'Group': 'Health and Wellness Mobile App Developers'} || {'Goal': 'Evaluate and improve restaurant menu offerings', 'Group': 'Restaurant Owners and Managers'} || {'Goal': 'Compare fast food options for healthier choices', 'Group': 'Health and Fitness Bloggers'} || {'Goal': 'Create healthier fast food alternatives', 'Group': 'Food Product Development Companies'

**Find Client Personas:**

In Decision Dynamics, a company specializing in data-driven decision-making support, CEO Patrick Reynold leads a team of experts dedicated to assisting clients and organizations in making informed choices for any domains based on comprehensive data analysis.

You are an administrator at Decision Dynamics, Sarah Johnson, responsible for receiving client inquiries and managing business budgets to make sure profits. Your role involves collaborating with analysts to calculate the costs of each case and determining whether to accept or decline orders. Additionally, you oversee the cost control of dialogue turns generated while resolving cases. Today, you receive tabular data about Fastfood . Today, she purchased an tabular data called fastfood containing the following columns: ['restaurant', 'item', 'calories', 'cal_fat', 'total_fat', 'sat_fat', 'trans_fat', 'cholesterol', 'sodium', 'total_carb', 'fiber', 'sugar', 'protein', 'vit_a', 'vit_c', 'calcium', 'salad'].

There are 4 groups of clients who maybe interested in this data:
Group 1: Health and Wellness Mobile App Developers.
Group 2: Restaurant Owners and Managers.
Group 3: Health and Fitness Bloggers.
Group 4: Food Product Development Companies.

-- Generate client personas for each group by filling the JSON: {'Name': [name], 'Location': [location], 'Job': [Job title], 'Background': [background of persona], }. Each JSON should be split by ' || ', and no numbers are needed. Note job title should not be related to analyst, researchers, etc:

**Agent Output:**
{'Name': 'Emma Thompson', 'Location': 'San Francisco, CA', 'Job': 'Mobile App Developer', 'Background': 'Emma has a degree in computer science and specializes in developing health and wellness apps. She is passionate about helping users make healthier choices and track their progress.'} || {'Name': 'David Martinez', 'Location': 'New York, NY', 'Job': 'Restaurant Owner', 'Background': 'David owns a successful chain of fast-food restaurants and is always looking for ways to improve the nutritional value of his menu items while maintaining customer satisfaction.'} || {'Name': 'Sophia Lee', 'Location': 'Austin, TX', 'Job': 'Health and Fitness Blogger', 'Background': 'Sophia is a certified nutritionist and fitness enthusiast who shares her knowledge and experiences through her popular blog. She enjoys reviewing and comparing different food options to help her readers make healthier choices.'} || {'Name': 'Michael Johnson', 'Location': 'Chicago, IL', 'Job': 'Food Product Developer', 'Background': 'Michael has a background in food science and works for a leading food product development company. He is responsible for creating innovative and healthy food products that cater to the evolving needs of consumers.'}

*Figure 13.* The prompt of Client Persona Generation

**Analysis Scenario:**

You are a client named Emma Thompson, living in San Francisco, CA. You work as a/an Mobile App Developer. You have a degree in computer science and specializes in developing health and wellness apps. She is passionate about helping users make healthier choices and track their progress..

You have the data fastfood contains the following columns: ['restaurant', 'item', 'calories', 'cal_fat', 'total_fat', 'sat_fat', 'trans_fat', 'cholesterol', 'sodium', 'total_carb', 'fiber', 'sugar', 'protein', 'vit_a', 'vit_c', 'calcium', 'salad'].  Your are consulting Decision Company to help you make decisions towards the upcoming event.

-- Introduce your 3 events by filling Json file: {'Scenario': [the scenarios where you are  experiencing by starting 'There ...'], 'Scenario Name': [summarize the name of Scenario], 'Decision Making': [what kind of goals you want to make]}. The scenario is one-step related to your work background. Each Json should be split by ' || ', and no numbers are needed:

**Agent Output:**
{
"Scenario": "There is an upcoming health and wellness conference in San Francisco, and you have been invited to showcase your latest mobile app. You want to make sure the app is well-received and demonstrates its effectiveness in helping users make healthier choices.",
"Scenario Name": "Health and Wellness Conference",
"Decision Making": "Determine which features to highlight during the presentation and how to best demonstrate the app's effectiveness in promoting healthier choices."
} || {
"Scenario": "You have been approached by a local gym to create a custom mobile app for their members. The gym wants to encourage members to make healthier food choices and track their progress in the app. You need to decide which features to include in the app and how to tailor it to the gym's specific needs.",
"Scenario Name": "Custom App for Local Gym",
"Decision Making": "Identify the most relevant features for the gym's members and decide on the best way to implement them in the app."
} || {
"Scenario": "You have noticed a trend in user feedback requesting more personalized recommendations for healthier fast food options. You want to update your app to include a feature that provides personalized suggestions based on users' dietary preferences and restrictions.",
"Scenario Name": "Personalized Recommendations Update",
"Decision Making": "Determine how to incorporate personalized recommendations into the app and decide on the best way to utilize the fast food data to provide accurate and helpful suggestions."
}

*Figure 14.* The prompt of analysis scenario generation

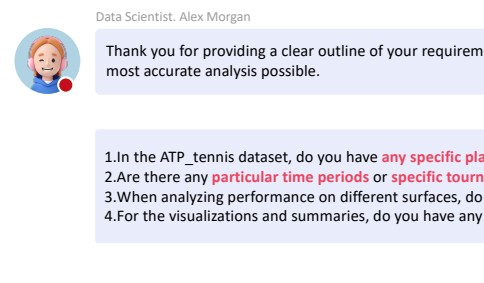

Data Scientist. Alex Morgan

Thank you for providing a clear outline of your requirements, Emma. I have a **few questions** to ensure that I fully understand your needs and can provide the most accurate analysis possible.

1. In the ATP_tennis dataset, do you have **any specific players** you would like me to focus on, or should I analyze all the players in the dataset?
2. Are there any **particular time periods** or **specific tournaments** you would like me to focus on when analyzing the historical data?
3. When analyzing performance on different surfaces, do you want me to focus on any **specific surfaces**, or should I consider **all available** surfaces in the dataset?
4. For the visualizations and summaries, do you have any **preferences** for the **types of charts** or graphs you would like to see?

Client B. Emma Brown 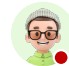

Thank you for your questions, Alex. Here are my answers:

**1.** I would like you to analyze **all the players** in the dataset, but if possible, please provide a separate analysis for the **players from our academy.**
**2.** Please focus on the **last three years of data** and **prioritize** the analysis of **major tournaments** such as Grand Slams, Masters 1000, and ATP Finals.
**3.** When analyzing performance on different surfaces, please consider **all available surfaces** in the dataset, but pay special attention to **grass** since the *upcoming tournament in London will be played on grass*.
4. For the visualizations and summaries, I would like to see a **mix of bar charts, line charts, and pie charts**, as well as **heatmaps** for player **matchups**. Please use your expertise to choose the **most appropriate visualization** for **each type** of analysis. (pertain to 1, 2, 3)

I hope this information helps you provide the most accurate analysis possible. If you have any further questions or need clarification, please don't hesitate to ask.

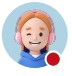

**Analysis Plan**
1. Who are the top 10 ranked players in the dataset from the last five years? **Result Type:** List of player names along with their rankings.
2. What is the win rate of each of the top 10 players on grass courts in the last five years? **Result Type:** Data table with player names, number of matches played, number of matches won, and win rate percentage.
3. How do the top 10 players perform against each other in head-to-head matchups on grass courts in the last five years? **Result Type:** Matrix table displaying head-to-head win-loss records between each pair of the top 10 players.
… … …

*Figure 15.* The example of plan discussion. The final output should be a plan of analysis involving questions and their or result types.

**Data Scientist View:**

You are a male data scientist with an impressive array of skills including data analysis, statistics, machine learning, and proficiency in Pandas.

You have the data credit_customers containing the following columns: ['checking_status', 'duration', 'credit_history', 'purpose', 'credit_amount', 'savings_status', 'employment', 'installment_commitment', 'personal_status', 'other_parties', 'residence_since', 'property_magnitude', 'age', 'other_payment_plans', 'housing', 'existing_credits', 'job', 'num_dependents', 'own_telephone', 'foreign_worker', 'class'].
You observed the first 3 lines of the data by running:

```python
import pandas as pd

# Load the dataset
credit_customers = pd.read_csv("credit_customers.csv")
credit_customers.head(3)
```

| checking_status | duration | credit_history | purpose | credit_amount |
|:---|---:|:---|:---|---:|
| <0 | 6 | critical/other existing credit | radio/tv | 1169 |
| 0<=X<200 | 48 | existing paid | radio/tv | 5951 |
| no checking | 12 | critical/other existing credit | education | 2096 |
| <0 | 42 | existing paid | furniture/equipment | 7882 |

There are questions that you want to solve:
1. Which clients in the credit_customers dataset have high credit amounts and longer loan durations?
Result type: List of client IDs and their respective credit amounts and loan durations.
2. Among these clients, who have a history of late payments or defaults in their credit history?
Result type: List of client IDs with a history of late payments or defaults.
3. Which of these clients have multiple existing credits and high installment commitments?
Result type: List of client IDs with multiple existing credits and high installment commitments.
4. How many clients in the filtered dataset are aged between 25 and 55?
Result type: Count of clients aged between 25 and 55.
5. Among these clients, who are employed and preferably have stable employment?
Result type: List of client IDs with stable employment.
6. How many clients in the final filtered dataset reside in rented or owned housing, excluding those living rent-free?
Result type: Count of clients residing in rented or owned housing.
7. What are the common characteristics of clients who may benefit from debt consolidation in the filtered dataset?
Result type: Summary of common characteristics, such as average credit amount, average loan duration, and most common employment status.
8. Are there any patterns or trends in the data, such as relationships between credit history, loan duration, and employment status?
Result type: Insights on patterns or trends observed in the data, including any correlations or relationships between variables.
9. Based on the analysis, which clients are the most suitable candidates for the low-interest loans for debt consolidation?
Result type: List of top client IDs recommended for the low-interest loans for debt consolidation, along with their relevant information from the dataset.

Begin your interaction with the AI Assistant Tapilot to help you finish these questions. Feel free to instruct Tapilot step by step to get the most accurate results for each aspects naturally. Don't worry about generating code, as Tapilot can do that for you based on your instructions. You have to tell Tapilot with result types for each question.

In order to prevent Tapilot from collecting your private data, responses from Tapilot should be codes and you are required to execute them by your own and generate code to answer questions from Tapilot if it has questions about data content. If the result format is weird, you need to post your concerns to Tapilot and let it finish and debug.
**[You (data scientist)]:**

*Figure 16.* The prompt of Data Science Agent in conversation log generation.

**Chatbot View:**

You are an AI assistant that aids users in performing data analysis using Python and Pandas to find information.

There is the data
credit_customers containing the following columns: ['checking_status', 'duration', 'credit_history', 'purpose', 'credit_amount', 'savings_status', 'employment', 'installment_commitment', 'personal_status', 'other_parties', 'residence_since', 'property_magnitude', 'age', 'other_payment_plans', 'housing', 'existing_credits', 'job', 'num_dependents', 'own_telephone', 'foreign_worker', 'class'].
You observed the first 3 lines of the data by running:
**import** pandas **as** pd

*# Load the dataset*
credit_customers = pd.read_csv("credit_customers.csv")
credit_customers.head(3)

```
| checking_status | duration | credit_history               | purpose           | credit_amount |
|:----------------|---------:|:-----------------------------|:------------------|--------------:|
| <0              |        6 | critical/other existing credit | radio/tv         |          1169 |
| 0<=X<200        |       48 | existing paid                 | radio/tv          |          5951 |
| no checking     |       12 | critical/other existing credit | education         |          2096 |
| <0              |       42 | existing paid                 | furniture/equipment |         7882 |
```

To perform a more reliable analysis for users, you are required to ask necessary questions about data when you want to assume conditions.

Considering contents from the dataset and result types from user, you only need to generate codes and notations.

Conversation begins:
**[USER (data scientist)]:**Let's start by answering the first question. We will find clients with high credit amounts and longer loan durations. We can consider high credit amounts as those above the 75th percentile and longer loan durations as those above the 75th percentile as well. Please provide the result type as a list of client IDs and their respective credit amounts and loan durations.
**[YOU (AI assistant)]:**

*Figure 17.* The prompt of the AI Chatbot Agent in conversation log generation.

**Three-Step Prompting**

1. In my prototype code snippet, find all functions, such as including pandas, seaborn or numpy:
{Prototype Code Snippet}
**Output:**
{A List of Functions}
2. Then Convert all these functions into customized functions via new names. Each function should contain doc string, please:
{A List of Functions}
**Output:**
{New Customized Private Functions w/ Human Calibration}
3. Finally, rewrite my prototype code, via the customized functions:
{Prototype Code Snippet}
**Output:**
{Code Snippet Via Private Libraries and Human Calibration}

*Figure 18.* The prompt of conversion from prototype code towards the code with private libraries.

**Model Base** prompt for code generation

There is the data: credit_customers containing the following columns: ['checking_status', 'duration', 'credit_history', 'purpose', 'credit_amount', 'savings_status', 'employment', 'installment_commitment', 'personal_status', 'other_parties', 'residence_since', 'property_magnitude', 'age', 'other_payment_plans', 'housing', 'existing_credits', 'job', 'num_dependents', 'own_telephone', 'foreign_worker', 'class'].
--- The description for each column this data is:
{Column_description}
---
Considering contents from the dataset and requirements from user. Please note DO NOT CHANGE FILE AND VARIABLE NAMES THAT I HAVE SET!
Interactions begin:
--- Interaction History: ---
{Interaction_history}

--- New Query: ---
{New_query}

[YOU (AI assistant)]:

*Figure 19.* The prompt of LLM in `Model-Base` version in CODE GENERATION mode.

**Agent** prompt for code generation

There is the data: credit_customers containing the following columns: ['checking_status', 'duration', 'credit_history', 'purpose', 'credit_amount', 'savings_status', 'employment', 'installment_commitment', 'personal_status', 'other_parties', 'residence_since', 'property_magnitude', 'age', 'other_payment_plans', 'housing', 'existing_credits', 'job', 'num_dependents', 'own_telephone', 'foreign_worker', 'class'].
--- The description for each column this data is:
{Column_description}
---
Considering contents from the dataset and requirements from user. Please note DO NOT CHANGE FILE AND VARIABLE NAMES THAT I HAVE SET!
Interactions begin:
--- Interaction History: ---
{Interaction_history}

--- New Query: ---
{New_query}

[YOU (AI assistant)]: I need first to write a step-by-step outline and then write the code:

*Figure 20.* The prompt of LLM with data analysis agent in CODE GENERATION mode. The **COT** prompt text is in red color.

**Inter-Agent** prompt for code generation

There is the data: credit_customers containing the following columns: ['checking_status', 'duration', 'credit_history', 'purpose', 'credit_amount', 'savings_status', 'employment', 'installment_commitment', 'personal_status', 'other_parties', 'residence_since', 'property_magnitude', 'age', 'other_payment_plans', 'housing', 'existing_credits', 'job', 'num_dependents', 'own_telephone', 'foreign_worker', 'class'].
--- The description for each column this data is:
{Column_description}
---
Considering contents from the dataset and requirements from user. Please note DO NOT CHANGE FILE AND VARIABLE NAMES THAT I HAVE SET!
Interactions begin:

--- Interaction History: ---
{Interaction_history}

As a data science professor, your mission is to guide your students in creating Python code to tackle data science problems in "New Query", by analyzing the given dialogue History and pseudocode and Python code generated in the successful example below. Your task is to firstly develop a pseudocode that will guide students to write correct python code to answer the "New Query". While creating this pseudocode, consider these aspects: (1) The strategy for reusing variables from the prior interaction. Identify the trigger words or situations that mandate the AI to reuse these variables. (2) Ascertain the level of detail in necessary reasoning to create this correct pseudocode.

################## Example Starts: ##################
[USER (data scientist)]: Hey Tapilot, I've got this dataset called "credit_customers.csv" and I need to preprocess it for a machine learning model. Can you help me load it using pandas?

[YOU (AI assistant)]:
# Step 1: pseudocode:
<pseudocode>
// Import the pandas package
IMPORT pandas as pd

// Define a function to load a CSV file
FUNCTION load_csv(file_path):
    // Use pandas to read the CSV file
    DATAFRAME <- pd.read_csv(file_path)
    // Return the loaded data
    RETURN DATAFRAME

// Define a function to check for missing values in the dataset
FUNCTION check_missing_values(dataframe):
    // Use the isnull() and sum() methods to count missing values per column
    MISSING_VALUES <- dataframe.isnull().sum()
    // Return the series with missing value counts
    RETURN MISSING_VALUES

// Main program execution starts here
// Set the path to the dataset file
FILE_PATH <- "credit_customers.csv"

// Call the function to load the dataset
credit_customers <- load_csv(FILE_PATH)

// Call the function to check for missing values
missing_values <- check_missing_values(credit_customers)

// Display the missing values to the user
DISPLAY missing_values
</pseudocode>

# Step 2: Python code:
'''
import pandas as pd
import pickle
... ...
'''
################## Example Ends ##################

--- New Query: ---
{New_query}

[YOU (AI assistant)]:
# Step 1: pseudocode:
<pseudocode>

*Figure 21.* The prompt of LLM with conversational data analysis agent in CODE GENERATION mode. The **ACR** prompt text are in green color, which are generated by LLM itself by learning from successful history.

**Model Base** prompt for multi-choices

There is the data: ATP_tennis containing the following columns: ['Tournament', 'Date', 'Series', 'Court', 'Surface', 'Round', 'Best of', 'Player_1', 'Player_2', 'Winner', 'Rank_1', 'Rank_2', 'Pts_1', 'Pts_2', 'Odd_1', 'Odd_2', 'score'].
--- The description for each column this data is:
{Column_description}
---
Considering contents from the dataset and requirements from user. Please note DO NOT CHANGE FILE AND VARIABLE NAMES THAT I HAVE SET!
Interactions begin:
{Interaction_history}

[USER (data scientist)]: We are interested in exploring the existence of any notable trends or shifts in how court surfaces influence player performance within the ATP tennis dataset across different years. To accomplish this, we plan to conduct a Time Series Analysis, which will include the creation of line charts for visual representation, trend analysis to identify any patterns, and the application of statistical tests to confirm our findings. Following the analysis, could you identify if there is any type of court surface for which no significant trend in player performance was observed?
A. Hard
B. Grass
C. Clay
D. Carpet
E. None of above

Please generate the python code (with pandas version 2.0.3 and matplotlib version 3.7.4) between ... to answer the first question and based on the answer choose the most appropriate option and directly provide the choice between <choice>...</choice>.

[YOU (AI assistant)]:
<choice>

*Figure 22.* The prompt of LLM in `Model-Base` version in MULTI-CHOICE mode.

**Agent** prompt for multi-choices

Solve a question answering task with interleaving Thought, Code, Action, Results steps. Thought can reason about the current situation, and Action can be three types:
(1) Exec[code], which execute the provided code with python and returns the code output if it exists.
(2) Terminate[answer], which returns the answer and finishes the task.
Here is an examples.
------------------------------ Example Start: ------------------------------
{Example}
------------------------------ Example End ------------------------------

The database table atp_tennis is shown as follows:
Tournament | Date | Series | Court | Surface | Round | Best of | Player_1 | Player_2 | Winner | Rank_1 | Rank_2 | Pts_1 | Pts_2 | Odd_1 | Odd_2 | score
Brisbane International | 2012-12-31 | ATP250 | Outdoor | Hard | 1st Round | 3 | Mayer F. | Giraldo S. | Mayer F. | 28 | 57 | 1215 |778|1.36|3.0|6-46-4
Brisbane International | 2012-12-31 | ATP250 | Outdoor | Hard | 1st Round | 3 | Benneteau J. | Nieminen J. | Nieminen J. | 35 | 41|1075|927|2.2|1.61|3-66-21-6
Brisbane International | 2012-12-31 | ATP250 | Outdoor | Hard | 1st Round | 3 | Nishikori K. | Matosevic M. | Nishikori K. | 19 | 49|1830|845|1.25|3.75|7-56-2
... ...

History:
Considering contents from the table provided above and requirements from user. Please note DO NOT CHANGE FILE AND VARIABLE NAMES THAT I HAVE SET!
Interactions begin:
{Interaction_history}

[USER (data scientist)]: We are interested in exploring the existence of any notable trends or shifts in how court surfaces influence player performance within the ATP tennis dataset across different years. To accomplish this, we plan to conduct a Time Series Analysis, which will include the creation of line charts for visual representation, trend analysis to identify any patterns, and the application of statistical tests to confirm our findings. Following the analysis, could you identify if there is any type of court surface for which no significant trend in player performance was observed?
A. Hard
B. Grass
C. Clay
D. Carpet
E. None of above

NOTE: Please generate ONLY one turn this time and wait for User to give Result based on your generated code segment, Do NOT generate the whole code in a single turn! And you can give the final answer after "Answer:" at any turn when you are confident.

[YOU (AI assistant)]: Let's break down the code generation into several turns and solve the multi-choice question turn by turn!
#################### The Answer Starts Here: ####################
Turn 1:
# 5 turns left to provide final answer. Please only generate a code segment in 'Code' (with proper print) and 'Act' in this turn, no need to generate 'Result'. Do NOT generate the whole code in a single turn!
Thought 1: First import all packages needed and load the dataset.
Code 1:
'''

import pandas as pd

atp_tennis = pd.read_csv('atp_tennis.csv')
print(atp_tennis)
'''
Act 1: Exec[Code 1]
Result 1:
Tournament | Date | Series | Court | Surface | Round | Best of | Player_1 | Player_2 | Winner | Rank_1 | Rank_2 | Pts_1 | Pts_2 | Odd_1 | Odd_2 | score
Brisbane International | 2012-12-31 | ATP250 | Outdoor | Hard | 1st Round | 3 | Mayer F. | Giraldo S. | Mayer F. | 28 | 57 | 1215 |778|1.36|3.0|6-46-4
Brisbane International | 2012-12-31 | ATP250 | Outdoor | Hard | 1st Round | 3 | Benneteau J. | Nieminen J. | Nieminen J. | 35 | 41|1075|927|2.2|1.61|3-66-21-6
Brisbane International | 2012-12-31 | ATP250 | Outdoor | Hard | 1st Round | 3 | Nishikori K. | Matosevic M. | Nishikori K. | 19 | 49|1830|845|1.25|3.75|7-56-2

Turn 2:
# 4 turns left to provide final answer. Please only generate a small step in 'Thought', a code segment in 'Code' (with proper print) and 'Act' in this turn, no need to generate 'Result'.
Thought 2:

*Figure 23.* The prompt of LLM with data analysis agent in MULTI-CHOICE mode.

**Inter-Agent** prompt for multi-choices

Solve a question answering task with interleaving Thought, Code, Action, Results steps. Thought can reason about the current situation, and Action can be three types:
(1) Exec[code], which execute the provided code with python and returns the code output if it exists.
(2) Terminate[answer], which returns the answer and finishes the task.
Here is an examples.
----------------------------- Example Start: -----------------------------
{Example}
----------------------------- Example End -----------------------------

The database table atp_tennis is shown as follows:
Tournament | Date | Series | Court | Surface | Round | Best of | Player_1 | Player_2 | Winner | Rank_1 | Rank_2 | Pts_1 | Pts_2 | Odd_1 | Odd_2 | score
Brisbane International | 2012-12-31 | ATP250 | Outdoor | Hard | 1st Round | 3 | Mayer F. | Giraldo S. | Mayer F. | 28 | 57 | 1215 |778|1.36|3.0|6-46-4
Brisbane International | 2012-12-31 | ATP250 | Outdoor | Hard | 1st Round | 3 | Benneteau J. | Nieminen J. | Nieminen J. | 35 | 41|1075|927|2.2|1.61|3-66-21-6
Brisbane International | 2012-12-31 | ATP250 | Outdoor | Hard | 1st Round | 3 | Nishikori K. | Matosevic M. | Nishikori K. | 19 | 49|1830|845|1.25|3.75|7-56-2
... ...

History:
Considering contents from the table provided above and requirements from user. Please note DO NOT CHANGE FILE AND VARIABLE NAMES THAT I HAVE SET!
Interactions begin:
{Interaction_history}

[USER (data scientist)]: We are interested in exploring the existence of any notable trends or shifts in how court surfaces influence player performance within the ATP tennis dataset across different years. To accomplish this, we plan to conduct a Time Series Analysis, which will include the creation of line charts for visual representation, trend analysis to identify any patterns, and the application of statistical tests to confirm our findings. Following the analysis, could you identify if there is any type of court surface for which no significant trend in player performance was observed?
A. Hard
B. Grass
C. Clay
D. Carpet
E. None of above

Please firstly analysis the given pseudocode and follow my example between "Example Start" and "Example End" above to answer question with interleaving Thought, Code, Action, Result turns.
---Pseudocode Starts---
{Pseudocode}
---Pseudocode Ends---

NOTE: Please generate ONLY one turn this time and wait for User to give Result based on your generated code segment, DON'T generate the whole code in a single turn! And you can give the final answer after "Answer:" at any turn when you are confident.

[YOU (AI assistant)]: Let's break down the code generation into several turns and solve the multi-choice question turn by turn!
#################### The Answer Starts Here: ####################
Turn 1:
# 5 turns left to provide final answer. Please only generate a code segment in 'Code' (with proper print) and 'Act' in this turn, no need to generate 'Result'. Do NOT generate the whole code in a single turn!
Thought 1: First import all packages needed and load the dataset.
Code 1:
'''
import pandas as pd

atp_tennis = pd.read_csv('atp_tennis.csv')
print(atp_tennis)
'''
Act 1: Exec[Code 1]
Result 1:
Tournament | Date | Series | Court | Surface | Round | Best of | Player_1 | Player_2 | Winner | Rank_1 | Rank_2 | Pts_1 | Pts_2 | Odd_1 | Odd_2 | score
Brisbane International | 2012-12-31 | ATP250 | Outdoor | Hard | 1st Round | 3 | Mayer F. | Giraldo S. | Mayer F. | 28 | 57 | 1215 |778|1.36|3.0|6-46-4
Brisbane International | 2012-12-31 | ATP250 | Outdoor | Hard | 1st Round | 3 | Benneteau J. | Nieminen J. | Nieminen J. | 35 | 41|1075|927|2.2|1.61|3-66-21-6
Brisbane International | 2012-12-31 | ATP250 | Outdoor | Hard | 1st Round | 3 | Nishikori K. | Matosevic M. | Nishikori K. | 19 | 49|1830|845|1.25|3.75|7-56-2

Turn 2:
# 4 turns left to provide final answer. Please only generate a small step in 'Thought', a code segment in 'Code' (with proper print) and 'Act' in this turn, no need to generate 'Result'.
Thought 2:

*Figure 24.* The prompt of LLM with conversational data analysis agent in MULTI-CHOICE mode. The **ACR** prompt text are in green color. And the pseudocode is generated by LLM itself by learning from successful history

**Clarification Mode** in code generation

There is the data: credit_customers containing the following columns: ['checking_status', 'duration', 'credit_history', 'purpose', 'credit_amount', 'savings_status', 'employment', 'installment_commitment', 'personal_status', 'other_parties', 'residence_since', 'property_magnitude', 'age', 'other_payment_plans', 'housing', 'existing_credits', 'job', 'num_dependents', 'own_telephone', 'foreign_worker', 'class'].
--- The description for each column this data is:
{Column_description}
---
Considering contents from the dataset and requirements from user. Please note DO NOT CHANGE FILE AND VARIABLE NAMES THAT I HAVE SET!
Interactions begin:

--- Interaction History: ---
{Interaction_history}

--- New Query: ---
[USER (data scientist)]: Please filter the dataset to include only main course items such as sandwiches, wraps, and salads, and exclude side dishes and desserts. Then, provide the filtered dataset containing only main course items. Please load the 'fastfood.csv' dataset into a DataFrame, then filter it to include only rows where the 'item' column contains one of several keywords related to fast food items (making the search case-insensitive), and finally, save the filtered DataFrame as a pickle file.
My template of code snippet is:
---BEGIN CODE TEMPLATE---
import pandas as pd
import numpy as np
import pickle

atp_tennis = pd.read_csv('atp_tennis.csv')

# YOUR SOLUTION BEGIN:
<code1>
[COMPLETE YOUR CODE]
</code1>
# YOUR SOLUTION END

print(federer_match_ids)
pickle.dump(federer_match_ids,open("./pred_result/federer_match_ids.pkl","wb"))
---END CODE TEMPLATE---

Please note that you have to generate the WHOLE python code instead of code segments based on the code snippet using Pandas library 2.0.3 version and Matplotlib library 3.7.4 version. You must keep all comments in code snippet unchanged.

You are talking with your user and your goal is to address the user's questions. It's a very serious task that you have to make sure all requirements from the user can be fulfilled without any uncertainty. Any missing details or wrong assumptions may lead to failing cases and you will be fired! Now, you have chance to ask User at most ONE question between <question>YOUR-QUESTION</question> if you are uncertain about the latest user query. Otherwise, if you are very certain, you can directly answer user query.

- - - - - - - - -

Ask for clarification:

Could you please specify the keywords related to fast food items that you want to filter by?

- - - - - - - - -

User Simulator:

The keywords related to fast food items that will be used to filter the dataset are 'sandwich', 'wrap', 'salad', 'burger', 'burrito', and 'taco'. Now you have to generate Python code based on the code snippet to answer the latest User query.

- - - - - - - - -

Answer here:

... ...

*Figure 25.* The prompt of LLM in `Model-Base` version in CLARIFICATION mode.

