# OpenReview forum: "Are Large Language Models Ready for Multi-Turn Tabular Data Analysis?"
_ICML.cc/2025/Conference — ICML 2025 poster_

### Official Review · Reviewer_Wqoz · 2025-03-06

**Overall Recommendation:** 4

**Summary:**

The authors create a synthetic conversational dataset with dialogues about the data tables. The authors scrape the tables from Kaggle. To generate the conversation, the authors organize a multi-agent multi-turn conversation where each agent is an LLM instance prompted to play a specific role.The authors then perform extensive human expert validation of the generated conversions and ground truth formulation as MCF or code. A thorough comparison to 8 competing datasets/benchmarks is performed. Having the dataset in place, the authors propose Adaptive Conversation Reflection (ACR) - an agentic setup that learns from the conversational dataset and improves the scores over the naked LLM and CoT-LLM.

**Claims And Evidence:**

All the claims of the paper are well supported.

1. The reasoning over the table data is an important field of study
2. The dataset creation procedure is rigorous enough. Even though the original dialogues are synthetic, they are well validated and curated by human experts.
3. Formal (objective) automated evaluation ground truth and metrics are worked out.
4. The feature coverage compared to the baseline benchmarks is rich.
5. ACR seems to improve the scores over a fairly strong baseline of LLM-CoT.

**Essential References Not Discussed:**

I am not aware of any essential literature that was missed.

**Ethical Review Concerns:**

What are the licenses of all the tables that you had harvested from Kaggle?

**Ethical Review Flag:**

Flag this paper for an ethics review.

**Ethics Expertise Needed:**

["Legal Compliance (e.g., GDPR, copyright, terms of use)"]

**Experimental Designs Or Analyses:**

Yes.

**Methods And Evaluation Criteria:**

Yes.

**Other Comments Or Suggestions:**

The citation of “Li et al., 2024a” is incorrect, should be “Li et al, 2023”.

The term “Action” is not very clear. The mentioned term “Clarification” scenario would be more suitable as the main name.

**Other Strengths And Weaknesses:**

The idea of having a “private” version of pandas API is interesting and helps to assess how well LLMs close the Open/Closed source domain gap.

**Questions For Authors:**

Writing could be slightly improved.

1. The name Decision Company is not clear, specifically, why “decision”?

2. The consistency of the terminology is not ideal. The authors write:

094  efficient creation of COTA

Whereas:

034 In this paper, we introduce COnversational Tabular data Analysis (COTA).

The derived phrase “creation of analysis” is not correct.

3. “Action mode” and “Action types” lack mathematical notation for them. How are they related to the notation of the Task Formulation section?

4. What is the definition of a “logic”? What are curly brackets in “Re-Org One-Shot Reasoning” section? (This is the most IMPORTANT question).

**Relation To Broader Scientific Literature:**

-

**Theoretical Claims:**

No theoretical claims.

---

> ### Author Rebuttal · Authors · 2025-03-31
>
> **Additional Evidence in Anonymous Link: https://anonymous.4open.science/r/additional_materials-E646**. We will use `B.X` to index evidence in the following rebuttal:
>
> **C1: The name Decision Company is not clear, specifically, why “decision”?"**
>
> **A1:** The name "DECISION COMPANY" refers to the multi-agent sandbox environment created by the authors to simulate realistic data analysis scenarios. Based on the paper, this environment is designed to support data-driven decision-making processes. The name likely reflects its purpose: facilitating decisions through data analysis in a company-like setting where different agents (Administrator, Client, Data Scientist, AI Chatbot) interact to answer analysis questions for decision making.
>
> **C2: Suggested Formula of Action Types**
>
> **A2:** Thank you for the suggestion. Given that our benchmark evaluates LLMs on separate actions, we initially didn't formulate each action individually. For more mathematical rigor, we can indeed extend our notation as follows:
>
> Let $s_i \in S$, where $s_i$ represents the selected action mode from an enumerated set of available actions listed in Section 2.
>
> The task formulation could then be improved as generating answers:
> $a_{it} = f_\theta(u_t, H, T, s_i)$
>
> This represents that given conversation history $H$, tabular data $T$, and the current query $u_t$, the agent's response $a_{it}$ is conditioned on the specific action mode $s_i$.
>
> **C3: What is the definition of a “logic”? What are curly brackets in “Re-Org One-Shot Reasoning” section? (This is the most IMPORTANT question).**
>
> **A3:** Thanks for this question.
>
> In Figure 11, "logic" represents the intermediate reasoning process that connects natural language queries to executable code or analytical answers. More formally, $m_{t-1}$ denotes the **inferred** reasoning pathway between a user query $u_{t-1}$ and the corresponding answer $a_{t-1}$ from the previous turn. This intermediate representation functions as pseudocode, which is a structured thought process that bridges natural language intent and formal execution.
>
> The curly brackets in the "Re-Org One-Shot Reasoning" section serve as an organizational construct. The notation $p_{t-1} = (u_{t-1}; \lbrace m_{t-1}; a_{t-1} \rbrace)$ mathematically represents our one-shot example structure, where:
> - $u_{t-1}$ is the previous user query
> - $\lbrace m_{t-1}; a_{t-1} \rbrace$ represents the pairing of inferred logic and answer. **In the prompt, this part is specifically highlighted with special symbols** to lead LLMs to follow this reasoning procedure.
>
> By structuring examples to show the reasoning process $m_{t-1}$ followed by its corresponding answer $a_{t-1}$, we enable the model to learn the pattern of first generating logical reasoning steps before producing final answers. This significantly improve performace of LLMs in conversational data analysis tasks in a simple manner.
>
> **C4: Data Resources & Dataset Liscense**
>
> **A4:** Thank you for this important question regarding our dataset resources and licensing.
>
> We have indeed provided comprehensive license information in our paper's Appendix B. Specifically, in Appendix B.1, we clearly state that the COTA dataset is available under the CC BY-SA 4.0 license (Creative Commons Attribution-ShareAlike 4.0 International).
>
> Regarding the source data, Appendix B.2 documents that all tabular data utilized in constructing COTA were obtained from Kaggle under either:
> 1) Public Domain Mark designation, or
> 2) CC BY (Creative Commons Attribution 4.0 International) licensing
>
> Also, we provide all detailed liscenses in `B.1` in the Anonymous Link. We would appreciate if you could go through it. Thanks.

---

### Official Review · Reviewer_K6vh · 2025-03-12

**Overall Recommendation:** 3

**Summary:**

This paper introduces CoTA, a benchmark to evaluate the effectiveness of LLMs in multi-turn conversational tabular data analysis scenarios. The authors' motivation is to address the lack of realistic, quantitative evaluation datasets by creating conversational data through an innovative multi-agent sandbox environment. CoTA includes diverse conversations scenarios to rigorously assess conversational abilities of LLMs. Additionally, the authors propose ACR, a self-generated reflection strategy to improve LLM performance, achieving notable enhancements over baseline approaches.

**Claims And Evidence:**

1. The authors argue that the ACR method significantly improves conversational agent performance, and empirically evidenced.

2. The authors claim that CoTA is a scalable and realistic benchmark for conversational data analysis. However, the authors may show its scalability in the aspect of the cost (since they use LLMs to generate them). In addition, while their purpose is really similar to text-to-SQL tasks, CoTA lacks comparison with them.

**Essential References Not Discussed:**

I think Spider 2.0 [1] should be discussed in the paper, since it is also a benchmark that tries to make realistic text-to-SQL scenarios where tabular data is really messy and the generated code (i.e., SQL) is really long.

[1] Spider 2.0: Evaluating Language Models on Real-World Enterprise Text-to-SQL Workflows, ICLR 2025.

**Experimental Designs Or Analyses:**

The authors tested several advanced LLMs, including Mistral, Llama, Claude, and GPT-families, across four conversation scenarios, i.e., Normal, Action, Private, and Private Action. They also performed extensive error analysis.

**Methods And Evaluation Criteria:**

1. They use Accuracy as metric, to measure correctness of code generation and answers.

**Other Comments Or Suggestions:**

See above weaknessess.

**Other Strengths And Weaknesses:**

**Strengths**
1. Comprehensive benchmark addressing realistic multi-turn conversational scenarios.

2. Effective use of a novel multi-agent sandbox for dataset creation.

3. Extensive experimental results, including multiple model comparisons and error analysis.

**Weaknesses**
1. Limited explicit discussion on scalability constraints or the cost implications of human-in-the-loop annotations for widespread practical use.

2. Relies heavily on GPT-4 based agents for conversation generation, possibly limiting diversity in generated data.

**Questions For Authors:**

1. What is the main differences of benchmark characteristics, compared to Spider 2.0.

2. Can you provide a used cost for generating conversations, and also used of human annotators?

3. Is it possible to generate conversations using open-sourced models like Llama?

4. What can be the practical applications of such conversational tabular analysis?

**Relation To Broader Scientific Literature:**

The literature is closely related on LLM-based code generation such as text-to-SQL.

**Theoretical Claims:**

As best as I know, there are no theoretical claims.

---

> ### Author Rebuttal · Authors · 2025-03-31
>
> **Paper Reference in: https://anonymous.4open.science/r/additional_materials-E646**.
>
> **C1: Comparison with Spider 2.0**
>
> **A1:** In summary, our benchmark differs from Spider 2.0 in several important aspects:
> - We focus on **conversational multi-turn** interactions for Python code, while Spider 2.0 primarily evaluates **single-turn** text-to-SQL capability;
> - Our benchmark includes more data analysis and scientific questions, including statistical and machine learning tasks such as clustering and linear regression;
> - Spider 2.0 emphasizes schema understanding and retrieval since their input data is larger (> 1000 columns), whereas our work prioritizes conversational compatibility and realistic data science problems. We will add these discussion in the Related Work section and Tab. 1 if fortunately given 1 additional page in camera-ready version. Thanks!
>
> **C2: Annotation Cost**
>
> Thanks for asking. Similar to Spider 2.0 and BigCodeBench for complex task annotation, we recognize annotators through authorship acknowledgment rather than direct financial compensation. Fortunately, to maintain data privacy and prevent GPT API abuse, annotators accessed our controlled annotation system, which documented a total working time of 3,677 minutes and accumulated GPT usage costs of  71.39 USD. While we did not provide monetary compensation to annotators, we can estimate the equivalent human expert cost by referencing established rates from research involving real data analysis experts [13-14], which indicates an average rate of 0.37875 USD per minute. Using this conversion, total human annotation would amount to 1,392.66 USD.
>
> Therefore, the comprehensive benchmark development cost totals 1,464.03 USD. This represents a cost-effective and scalable approach, with each data point in COTA costing approximately **1.45 USD**, significantly more economical than comparable benchmarks such as BIRD-SQL (**6.13 USD**) and TableBench (**6 USD**), despite COTA's increased complexity and deeper expertise.
>
> **C3: Is it possible to generate conversations using open-source models like Llama?**
>
> **A3:** We appreciate this question. Currently, we found that weaker LLMs struggle with this task. When we initially experimented with GPT-3.5-Turbo (the most popular LLM at that time), it exhibited significant hallucination issues after 2.5 conversation turns on average and failed to follow the Analysis Plan generated in Section 3.1. This required more expert workflow even than was generated totally by themselves.
>
> Therefore, we utilized GPT-4, the most capable model available when starting this project, as base model for sandbox construction. As shown in Figure 3, our sandbox based on this can support very long-turn conversations (14.15), similar to realistic complex task-solving scenarios. External human evaluations in Section 5 confirm the quality of these conversations.
>
> Recently, we observed that newer models like Llama 3.3 70B demonstrate improved capabilities for high-quality data annotation and data analysis questions, as shown in Table 3. This model could potentially serve as a replacement for GPT-4 in future studies. Thanks for your suggestion.
>
> **C4: Potential Limited Diversity**
>
> **A4:** The main reason why we use GPT-4 is its broad and diverse knowledge due to intensive pre-training. In the paper, we show scenario diversity evaluation in Tab.2 of **Section 5** and detailed fine-grained diversity analysis in **Appendix P.2** in terms of domain, result type, action, query, package, which actually shows our work already covers most comprehensive aspects of data analysis compared to related works as far as we know.
>
> **C5: What are the practical applications of conversational tabular analysis?**
>
> **A5:** Tabular data analysis is ubiquitous in daily operations.  Automatic tabular data analysis can help users make informed decisions through natural language interactions, without requiring specialized skills in complex tabular data understanding, coding or even domain knowledge, which also can improve efficiency for data scientists or relevant users.
>
> As we stated in L 23-32, users rarely express their intentions completely in a single turn and often have follow-up questions based on previous responses or other actions (as we summarized and listed in the paper). Therefore, conversational capability is necessary for complex tasks. All Leading AI assistants like ChatGPT, Claude, and DeepSeek support multi-turn interaction because conversation is the most natural form of human communication. Therefore, conversational tabular data analysis can accelerate and improve decision making for finance, health care, policy making, which usually store their records and data in more structrued format.
>
> In this case, a comprehensive, scalable (to prevent data leakage) benchmark should be proposed to make users understand the capabilities of LLMs.

---

> > ### Comment · Reviewer_K6vh · 2025-04-03
> >
> > Thank you for the rebuttal; I have raised my score to 3. I strongly recommend that the authors incorporate the discussion, especially regarding annotation costs and the use of other LLMs like Llama or GPT-4.

---

> > > ### Author Response · Authors · 2025-04-04
> > >
> > > Dear Reviewer,
> > >
> > > Thank you for reading our paper and our rebuttal. We will incorporate both the annotation cost analysis and discussion about LLM usage into the paper.  Your comments and suggestions make our work more rigorous. Thanks!
> > >
> > > Best,

---

### Official Review · Reviewer_xy4y · 2025-03-12

**Overall Recommendation:** 2

**Summary:**

# Summary

This paper constructs a novel benchmark for the task of "conversational tabular data analysis." The benchmark is constructed using a complex process of interacting agents, aided with human annotation, starting from a set of 5 "data sources" and 18 "topics". This ultimately yields a benchmark of approximately 1k examples, over which the authors conduct a benchmarking study.

This is a very important area of research and it requires both rigorous empirical studies, and reliable high-quality benchmarks; the current study is an attempt at both. However, I have some concerns about the current study. The dataset creation process is extremely complex. This has two consequences: (1) it is difficult for the authors to describe it with sufficient detail and clarity; (2) it makes it difficult to assess whether their design choices (of which there are many) affect the results of the benchmark. I also have some concerns about the value of the empirical insights gleaned from this study. Overall, I think that this is an important direction, but that the paper is not ready for acceptance in its current form.

# Major comments

* The dataset design process is almost incomprehensibly complex, and it is made even more difficult to understand by the writing in the paper. The are so many points that aren't clearly described; here are just a sample:
- The authors state (L78) that "multiple choice questions" are a response type. How can this be evaluated, and how can all responses be distilled into either code generation or multiple choice types? What is an example of this kind of response? This is not clear.
- The authors introduce 6 action types, saying simply that "we identify 6 common actions during conversations". How are these identified? How do we know that these represent a complete action space for conversational tabular data analysis tasks? These actions are also not clearly described; for example they define the Plot_QA action by simply saying "The Plot_QA action helps users understand plot-derived insights" -- but this is not a definition.

* The data used as the basis for the benchmark are not clearly described. Relegated to the Appendix, in B.3 and Figure 9(a), we see that there are "18 topics and 5 sources of COTA". This still does not answer basic questions, such as: how many tables comprise the benchmark, and exactly which ones? What is the difference between a table and a "source"? This is critical, fundamental information about the benchmark that is not clearly provided in the paper, and makes the results nearly impossible to reliably assess.

* Simple procedures, such as evaluation, are also not clearly described. For example, the paper says that "Each question is provided by an expected result type, such as dataframes, lists, or various plot types" but does not enumerate the expected result types. Similarly, the evaluation metrics are not clearly defined. For example, the definition of AccR, one of the core evaluation metrics, is simply limited to "we extend Acc to include a recall-based adjustment for instances involving private libraries" which does not give any details about the metric.

* Collectively, the above issues make the empirical results nearly impossible to assess. Furthermore, the authors' evaluation of the existing empirical results is quite limited and does not seem to lead to new insights about how to improve models.

# Minor comments

* The paper relies a lot on annotations by the authors. For example, the authors do the human-sandbox annotation in Section 2, and the error analysis in 7. It would be more reliable (less prone to bias) to have external annotators conduct these annotation steps.

* Why is only GPT-4-32k shown in Figure 7? It is not the best-performing model, and there isn't any other clear differentiator that makes this model particularly interesting to plot versus the others.

# Typos etc.

* "Conversational Tabular Data Analsis" in abstract does not need to be capitalized.
* Section 1: "Among the vast types of data available, tabular data stands out as one of the most prevalent and interpretable formats organized by rows and columns" -- isn't tabular data the only format organized by rows and columns, by definition?
* In several places the paper uses the word "codes" where the authors seem to mean "code" (as in, Python code).

**Claims And Evidence:**

See above.

**Essential References Not Discussed:**

See above. I would suggest additional references to existing works that apply language models to tabular data tasks (of which there are several).

**Experimental Designs Or Analyses:**

See above.

**Methods And Evaluation Criteria:**

See above.

**Other Comments Or Suggestions:**

See above.

**Other Strengths And Weaknesses:**

See above.

**Questions For Authors:**

See above.

**Relation To Broader Scientific Literature:**

See above.

**Theoretical Claims:**

See above.

---

> ### Author Rebuttal · Authors · 2025-03-31
>
> **Additional Evidence & Paper Reference in: https://anonymous.4open.science/r/additional_materials-E646**. We will use `B.X` to index evidence in the following rebuttal:
>
> **C1: Why the annotation is complex and annotators are also authors?**
>
> **A1:**
> Because the task that we are researching is complex. Unlike traditional NLP or mathematical tasks that rely on common knowledge or basic skills, our benchmark demands specialized expertise in tabular data analysis, statistical reasoning, coding proficiency, and domain experience. These requirements make conventional crowdsourcing approaches impractical for several reasons:
>
> - The technical depth required exceeds what typical annotation platforms can support.
> - The extended timeline is necessary for comprehensive task development and refinement.
> - The iterative workflow requires continuous expert feedback and adaptation.
>
> This is also reflected in very recent related work: BIGCODEBench and SPIDER 2.0 (ICLR 2025) as Reviewer K6vh mentioned. These projects similarly implemented sophisticated workflows for their tasks and recognized substantial annotator contributions through authorship.
>
> Our benchmark presents greater complexity than previous efforts. Therefore, our benchmarks deserve more sophisticated method to construct and authorship for annotators for their intellectual contributions, which is also trending of complex task benchmarking.
>
> **C2: Code Generation & Multi-Choice Annotation:**
>
> **A2:** In COTA, we deliberately designed two distinct answer types to enable objective and consistent evaluation:
>
> - Code Generation: This involves LLMs generating Python code to analyze tabular data. This code is evaluated through execution-based test case scripts.
> - Multiple-Choice Questions: After code execution, users often need to interpret results and make decisions. Rather than accepting free-form text analysis, which is difficult to evaluate objectively, we convert analytical questions into multiple-choice format.
>
> For example, after analyzing ATP tennis data, instead of asking "What trends do you see in player performance?" (subjective, and require LLM-as-Judge to evaluate), we frame it as:
> ```
> Which surface shows no significant performance trend over time?
> A. Hard
> B. Grass
> C. Clay
> D. Carpet
> E. None of above
> ```
> The evaluation process of multi-choice is straightforward - we compare the model's selected option against the ground truth answer. Each multiple-choice question has a single correct answer determined during dataset construction and verified by our expert annotators. This can eliminate ambiguity in subjective assessment, which may introduce evaluation bias.
>
> **C3: 6 action annotation and plot_QA**
>
> **A3:** Our identification of these 6 common actions stems from multiple rigorous sources of evidence rather than mere assertion. As we discussed in L187-188, annotators summarized actions and inject into conversations by two resources:
> - The annotators have> 10-year experience of data analysis (L135), ensuring strong expertise. This expertise is further validated by our high initial inter-annotator agreement rate (detailed in line 217). Also human expert (**outside annotator**) evaluation in Section 5 (**Action-wise Metrics**) shows its high Action Commonness. Detailed evaluation script in Appendix P.
> - Also, the action types were derived through systematic analysis of prior literature and empirical observation. Each action is grounded in but not limited to established research:
>
> **Update Code (for debugging)**: given code generation is crictical in data analysis [1, 2, 3].
> **Fast Fail**: See Appendix J, [4, 8]
> **Clarification:** [6-9]
> **Best Guess:** [4, 5]
> **PlotQA:** Chart/Plot is one of most common data format during data analysis [1, 9].
>
> **C4: Data Source Description**
>
> **A4:** Please see Tab. 1-5 of `B.1` in our anonymous link, it contains 45 large professional tables (> 5k rows, > 40 cols on avg), the "source" means domains.
>
> **C5: Details about metrics and Result Type:**
>
> **A5:**
> - Evaluation Metrics: We will feel grateful if you cloud give a look at Appendix K, M
> - Result Type Enumeration: We have catalogued all result types and their distributional characteristics in Figure 13(b) of Appendix K. To improve navigability, we will add direct cross-references in the revised manuscript. Thank you.
>
> **C6: Why is only GPT-4-32k shown in Figure 7?**
>
> **A6:**
> The visualization demonstrates how the Inter-Agent approach with ACR (light blue) generally outperforms both the base model and standard Agent configurations across most categories. GPT-4-32k likely serves as this case study because it represents a middle-ground performer (and one of most available resources). We have results on Claude-3.5-Sonnet which contains similar findings in Tab. 6 in `B.2`.
>
> **C7: More Reference**
>
> **A7**. Due to page limit, we have included a more comprehensive literature review in Appendix O. We will expand Section 8 in the camera-ready version if accepted.

---

### Official Review · Reviewer_ez8j · 2025-03-12

**Overall Recommendation:** 3

**Summary:**

This paper proposed a benchmark, namely COTA, and a multi-agent environment named Decision Company to evaluate the performance of  LLMs in the task of Tabular generation. The paper used the proposed benchmark and multi-agent environment to evaluate the performance of  8 LLMs in the task of tabular generation.

I have concerns in Client Persona Generation. In this step, the 'Persona' are generated by actually prompt engineering using a LLM. Although expert supervision is involoved, the output generated from LLMs may still differ from the real human. Moreover, the agents are given certain information of a faked person, which may not be representative to all the professionals in that task. The authors may consider use RLHF to enable the agents to mimic the professionals in a better way.

Also, the proposed benchmark does not include other fields which maybe more challenging, such as medical, marting, and supply chain.

**Claims And Evidence:**

Yes.

**Essential References Not Discussed:**

No.

**Experimental Designs Or Analyses:**

Yes. I checked the validity of the experimental design.

**Methods And Evaluation Criteria:**

Yes.

**Other Comments Or Suggestions:**

No.

**Other Strengths And Weaknesses:**

Consider to enhance the Persona Generation step. Also consider to include more domains.

**Questions For Authors:**

The Claude-3.5-Sonnet outperforms other models according to Figure 6. Is this the same case when using other dataset as shown in Table 1?

**Relation To Broader Scientific Literature:**

This paper contributes to the task of Tabular data generation. It provides a dataset and methodology to evaluate the Tabular generation.

**Theoretical Claims:**

Do not apply.

---

> ### Author Rebuttal · Authors · 2025-03-31
>
> **Concern 1:Client Persona Generation**
>
> **Ans:** Thank you for this valuable feedback. While LLMs were utilized in persona generation, we implemented a rigorous multi-stage validation process specifically to address potential discrepancies between LLM-generated content and real human behavior. Our approach includes:
>
> Expert supervision at every stage of persona development, where data analysis professionals with 10+ years of experience reviewed and modified the personas based on their real-world client interactions.
> High inter-annotator agreement (92.78%), suggesting strong consistency in the evaluation of these personas by multiple domain experts.
> Comprehensive human evaluation on multiple metrics, including Scenario Diversity and Reasonableness (0.96), Conversation Topic Coherence (0.93), and Conversation Naturalness (0.95), demonstrating the real-world applicability of these personas.
>
> Furthermore, our human-in-the-loop approach was intentionally designed as a practical middle ground between purely synthetic data generation and expensive expert crowdsourcing.
>
> **Concern 2: Persona Representativeness**
>
> **Ans:** We appreciate the suggestion regarding RLHF for better mimicking professionals. Our current approach prioritizes diversity and domain expertise. The personas were intentionally designed to cover a wide range of domain-relevant scenarios (18 analysis topics across 5 common domains) with varied backgrounds and needs.
> Actually, the proposed DECISION COMPANY framework already incorporates expert feedback loops that serve a similar purpose in guiding agent behavior toward realistic professional practices. Our evaluation with 10 data analysis experts outside the author team validated that the scenarios and interactions were highly representative of real-world data analysis tasks.
> For future work, we are exploring how RLHF techniques could further enhance the quality of agent-based simulations within our framework, especially for extending COTA to additional domains.
>
> **Concern 3: Domain Coverage**
>
> **Ans 3:** Thanks for suggestion. Our current focus on financial, sports, food, consumer electronics, and housing was intentional for three key reasons:
>
> - These domains feature widely available open-source tabular datasets with minimal privacy and ethical concerns, enabling broad accessibility of the benchmark.
> - These domains represent common data analysis scenarios that don't require highly specialized domain expertise to validate, ensuring reliable quality assessment.
> - Additionally, our current data already covers advanced topics such as Healthcare (Food),  Financial (Credit Card, Bank) shown in Fig. 9.
>
> We view COTA as a foundation that can be expanded to more challenging domains. The evaluation framework, conversation action types, and metrics we've developed will transfer well to specialized domains in future extensions of this work.
>
> **Question:**
>
> **Ans 4:** Thank for asking this. Due to limited resource at this time, we test the performance of Claude-3.5-Sonnect against GPT-4 and CodeLlama on CoSQL a conversational text-to-SQL benchmark:
>
> Performance (EX) comparison on other datasets.
> | Model Name | CoSQL |
> |----------|----------|
> | CodeLlama | 35.7 |
> | GPT-4 | 68.2 |
> | Claude-3.5-Sonnet | 58.6 |
>
> From this table, we can observe that Claude-3.5-Sonnet didn't outperform GPT-4, which means different advanced models maybe strong in different programming languages due to different training corpus. In our Python-based data analysis code generation, claude-3.5-sonnet is the strongest model.

---

> > ### Comment · Reviewer_ez8j · 2025-04-03
> >
> > I apperciate that the authors provide explanations and extra experiments. I will keep my original rating.

---

> > > ### Author Response · Authors · 2025-04-04
> > >
> > > Thanks for reading our responses and acknowledgment. We do appreciate your suggestions and time in reading our work in details.
> > >
> > > Best,

---

### Decision · Program_Chairs · 2025-05-01

**Decision:**

Accept (poster)

**Comment:**

This paper proposes a synthetic conversational tabular data benchmark. While I understand the strengths of the paper, the reviewers mention few weaknesses, such as complex setup, unnatural data creation, and presentation issues. As is, this makes a weak accept.